# CUPS: Improving Human Pose-Shape Estimators with Conformalized Deep Uncertainty

**Harry Zhang** [1]   **Luca Carlone** [1]

## Abstract

We introduce CUPS, a novel method for learning sequence-to-sequence 3D human shapes and poses from RGB videos with uncertainty quantification. To improve on top of prior work, we develop a method to generate and score multiple hypotheses during training, effectively integrating uncertainty quantification into the learning process. This process results in a deep uncertainty function that is trained end-to-end with the 3D pose estimator. Post-training, the learned deep uncertainty model is used as the conformity score, which can be used to calibrate a conformal predictor in order to assess the quality of the output prediction. Since the data in human pose-shape learning is not fully exchangeable, we also present two practical bounds for the coverage gap in conformal prediction, developing theoretical backing for the uncertainty bound of our model. Our results indicate that by taking advantage of deep uncertainty with conformal prediction, our method achieves state-of-the-art performance across various metrics and datasets while inheriting the guarantees of conformal prediction. 3D visualization, code, and data will be available at **this website**.

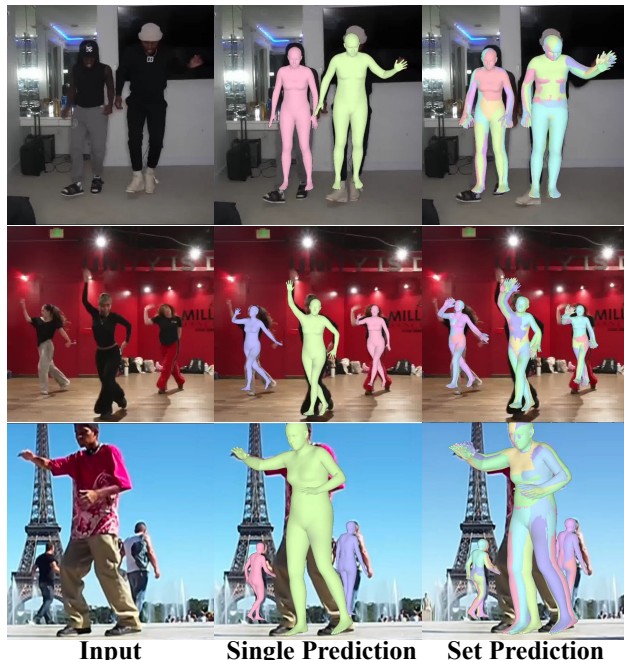

**Input**   **Single Prediction**   **Set Prediction**

Figure 1: CUPS sample results obtained on in-the-wild videos. Given RGB frames, CUPS reconstructs a sequence of 3D meshes, and then a conformal predictor calibrated using a deep uncertainty function —trained end-to-end with the human pose-shape estimator— quantifies the uncertainty of the output SMPL.

## 1. Introduction

Recovering a sequence of human meshes (*i.e.*, shapes and poses) from a monocular video is a fundamental challenge with wide-ranging applications in computer vision, robotics, AR/VR, and computer graphics. Such technology has the potential to minimize reliance on traditional motion capture systems or labor-intensive 3D annotations, facilitating the generation of human motion templates for tasks such as animating 3D avatars. The emergence of parametrized human models, such as SMPL (Kanazawa et al., 2018), which represent human body shape and pose with well-defined joint and structure parameters, made it possible for modern deep learning models to efficiently learn to predict human poses and shapes in a systematic way by directly regressing SMPL parameters from video inputs. At the same time, safety-critical applications, including robotics and autonomous vehicles, demand computer vision algorithms that are able to quantify the uncertainty in their estimates and possibly provide performance guarantees (Yang & Pavone, 2023).

Few existing works in human reconstruction have explored the direction of uncertainty-aware human pose and shape prediction due to two challenges. First, it is difficult to ensure the predicted pose and shape are close to the groundtruth under out-of-distribution data or heavy occlu-

[1]Massachusetts Institute of Technology. Correspondence to: Harry Zhang <harryz@mit.edu>.

*Proceedings of the $42^{nd}$ International Conference on Machine Learning*, Vancouver, Canada. PMLR 267, 2025. Copyright 2025 by the author(s).

sions. Second, an efficient human shape and pose prediction model takes video frames as input, where the data is not fully exchangeable. Such non-exchangeability makes it difficult for uncertainty quantification methods such as conformal prediction (Angelopoulos & Bates, 2021; Shafer & Vovk, 2008) to provide a formal statistical error bound between the estimation and the groundtruth. While a more recent line of work focuses on learning multiple outputs or learning variances as uncertainty (Zhang & Carlone, 2024), none has addressed the problem of providing a reliable statistical error bound when the data is not exchangeable. For practical uses in safety-critical scenarios, one should be able to tell when to *trust* the human reconstruction model.

To counter the aforementioned challenges, we first take inspiration from an important tool in statistical learning, conformal prediction (CP) (Shafer & Vovk, 2008; Angelopoulos & Bates, 2021), which uses a *post-training* calibration step to guarantee a user-specified coverage. Assume that we want to predict an output Y (*e.g.*, the true pose and shape of a human) from inputs X (*e.g.*, a sequence of frames). By allowing to predict *confidence sets* $C(X)$ (*e.g.*, a set of human reconstructions), CP guarantees the true value $Y$ to be included in $C(X)$ with confidence level $\alpha$, *i.e.*, $P(Y \in C(X)) \geq 1 - \alpha$, given a set of calibration examples $(X_i, Y_i) \in I_{\text{cal}}$ that are *exchangeable* with the test distribution. There are typically two steps involved in CP. In the calibration step, the conformity scores of the examples in the calibration set are ranked to determine a cutoff threshold $\mathbb{Q}_{1-\alpha}$, via quantile computation. In the prediction step, the conformity score measures the conformity between the output and the unknown ground-truth value, which is used —in conjunction with the threshold $\mathbb{Q}_{1-\alpha}$— to construct the confidence sets $C(X)$. By construction, the set $C(X)$ provides a quantification of uncertainty: each prediction in $C(X)$ has a conformity score above $(1-\alpha)$th quantile and hence is a plausible estimate for Y.

While CP is a flexible tool that can be applied to any machine learning model, it assumes the data to be *exchangeable*, *i.e.*, that the dataset distribution remains invariant under permutation. Such an assumption breaks in many real-world applications. For example, if the dataset comes from video frames, the constructed dataset is obviously not exchangeable since permuting sequences of frames changes the underlying distribution. While there have been techniques that aim to increase the exchangeability of video data by taking long video sequences (Zhang & Carlone, 2024) or observations from evenly-spaced cameras (Yang & Pavone, 2023), the theoretical guarantee of CP cannot be fully justified if the data is not exchangeable. To cope with the lack of exchangeability, we leverage a recent extension by Barber et al. (2023) that allows dealing with datasets where the exchengeability assumption no longer holds. The key idea in Barber et al. (2023) is to use

weighted quantiles to tackle data distribution shifts.

To bring CP into human reconstruction, we design a methodology that learns a *deep uncertainty score* of human reconstruction output in an end-to-end manner by predicting multiple hypotheses of human shapes and poses during training. The deep uncertainty value is then used in the calibration step by incorporating the theoretical toolkit provided in (Barber et al., 2023), retaining statistical guarantees even when exchangeability is violated. Our results indicate that taking advantage of deep uncertainty with conformal prediction, our method achieves state-of-the-art performance across various metrics and datasets. Using the probabilistic guarantee of correctness inherited from CP, we also provide theoretical lower bounds of performance for human mesh reconstruction when the data is not exchangeable. The result is **CUPS**, a **C**onformalized **U**ncertainty-aware human **P**ose-**S**hape estimator. To summarize, our contributions include:

- An uncertainty-aware 3D human shape-pose estimator from 2D RGB videos (Section 4.1 & Section 4.2).
- A novel method to conformalize 3D human estimates during training by learning a score function to rank the uncertainty of the proposed estimates (Section 4.2).
- A novel uncertainty quantifier for human reconstruction outputs using non-exchangeable conformal prediction and deep uncertainty function (Section 4.3).
- A theoretical analysis of the uncertainty when the data is not fully exchangeable and two practical lower bounds —one completely new and one adapted from (Barber et al., 2023)— for the coverage performance (Section 4.3).
- Quantitative and qualitative results that demonstrate the state-of-the-art results of our method on a variety of real-world datasets (Section 5).

## 2. Related Work

**3D Human Shape and Pose Estimation.** End-to-end approaches for human pose estimation include (Pavlakos et al., 2017; Sun et al., 2018). With the maturity of 2D human keypoints detection (Ho et al., 2022; Ma et al., 2022), more robust approaches focus on lifting 2D keypoints to 3D, resulting in better performance (Xu & Takano, 2021; Ma et al., 2021; Ci et al., 2019). In this scheme, deterministic methods learn to predict one single 3D output from the 2D input (Zhan et al., 2022; Zhang et al., 2022). In many applications, it is desirable to also recover the *shape* of humans beyond a skeleton of keypoints. (Loper et al., 2023; Kanazawa et al., 2018) propose SMPL, a universal parametrization for human pose and shape. MEVA (Luo et al., 2020) utilizes a VAE to encode the motion sequence and generate coarse human mesh sequences which are then refined via a residual correction. VIBE (Kocabas

et al., 2020), TCMR (Choi et al., 2021) and MPS-Net (Wei et al., 2022) encode representations of three different input lengths and then learns the mid-frame of the sequence with either a recurrent network or an attention module. GLoT (Shen et al., 2023) is a new model that decouples long-term and short-term correlations. We incorporate human mesh with uncertainty learning during training, forcing the network to output higher-quality meshes.

**Uncertainty in 3D Human Reconstruction.** Due to uncertainty such as occlusion in RGB inputs, deep generative models have been used in modeling conditional distributions for such problems. Mixed-density network (Li & Lee, 2019), VAE (Sharma et al., 2019), normalizing flows (Wehrbein et al., 2021), GAN (Li & Lee, 2019), and Diffusion models (Holmquist & Wandt, 2023; Shan et al., 2023) have all been applied to modeling such conditional distribution. Dwivedi et al. (2024) learn an explicit confidence value for occlusions. Motion-based methods such as (Zhang et al., 2024; Rempe et al., 2021) use physical contacts and trajectory consistency to make the uncertain estimates more robust. Zhang et al. (2023) use explicit anatomy constraints to improve model performance. Lastly, Biggs et al. (2020) generate a fixed number of hypotheses and learn to choose the best one. We learn an uncertainty score by augmenting training outputs and use the uncertainty score to probabilistically certify the outputs.

**Conformal Prediction.** CP is a powerful and flexible distribution-free uncertainty quantification technique that can be applied to any machine learning model (Angelopoulos & Bates, 2021; Shafer & Vovk, 2008) under the assumption of exchangeable data. Assuming the exchangeability of the calibration data, CP has desirable coverage guarantees. Thus, it has been applied to many fields such as robotics (Sun et al., 2024), pose estimation (Yang & Pavone, 2023), and image regression (Angelopoulos et al., 2022). More sophisticated CP paradigms have been also proposed to tackle distribution shift and online learning problems (Angelopoulos et al., 2024). More recently, theoretical grounding for conformal prediction beyond exchangeability assumption (Barber et al., 2023) has been proposed, which provides analysis tools for non-fully-exchangeable datasets such as videos in ML problems.

## 3. Problem Formulation

We are interested in the problem of learning a sequence of 3D human shapes and poses from a sequence of 2D RGB images. Formally, given the input 2D video sequence $\boldsymbol{X} \in \mathbb{R}^{H \times W \times 3 \times T}$, where $H, W$ are the dimension of each frame and $T$ is the length of the input sequence, our goal is to learn to output human shapes and poses, described by SMPL parameters $\boldsymbol{Y} := \{\boldsymbol{\theta}, \boldsymbol{\beta}\}$, where $\boldsymbol{\theta} \in \mathbb{R}^{24 \times 6 \times T}$ and $\boldsymbol{\beta} \in \mathbb{R}^{10 \times T}$ model the joint 6D pose and mesh shape, re-

spectively. We wish to learn a human reconstruction function $f_\theta(\boldsymbol{X})$ that approximates $\boldsymbol{Y}$. We are also interested in learning a deep uncertainty function $S_\theta(\boldsymbol{X}, \boldsymbol{Y})$, which measures the inherent uncertainty of the human reconstruction function $f_\theta$ when taking as input $\boldsymbol{X}$ and outputting $\boldsymbol{Y}$. Such a learned uncertainty function will be used as a conformity score in our method, and enables the construction of a prediction set via conformal prediction.

## 4. Method

We use a transformer-based architecture (Shen et al., 2023) to predict SMPL parameters from 2D video sequences. We also learn an uncertainty scoring model together with the reconstruction model. At test time, we use the uncertainty scoring model for conformal prediction. This results in CUPS, a Conformalized Uncertainty-aware human Pose-Shape estimator. The pipeline of CUPS is shown in Figure 2.

### 4.1. GLoT Human Reconstruction Model

The human reconstruction model in CUPS is based on the Global-to-Local Transformer (GLoT) architecture proposed by (Shen et al., 2023), which robustly leverages information learned with deep networks as well as human prior structures while decoupling the short-term and long-term modeling processes. We summarize the key components proposed in (Shen et al., 2023) below.

**Global Motion Modeling.** First, a pretrained ResNet-50 extracts features from individual frames, resulting in static tokens referred to as $\mathcal{S} = \{\boldsymbol{s}_1, \cdots, \boldsymbol{s}_T\} \in \mathbb{R}^{T \times 2048}$. The global motion modeling step begins by randomly masking a subset of static tokens along the temporal dimension, denoted as $\mathcal{S}^g \in \mathbb{R}^{(1-p)T \times 2048}$, where $p$ represents the mask ratio. The unmasked tokens are then passed through a global encoder. During the global decoder phase, the mean SMPL parameters encoded by an MLP (SMPL tokens) are padded into the masked positions and the entire sequence is fed into the global decoder, which generates a long-term representation. Subsequently, the global motion modeling step applies an iterative regressor (Kanazawa et al., 2018; Kocabas et al., 2020) to obtain the global initial SMPL sequence, denoted by $\boldsymbol{Y}^g = \{\boldsymbol{\theta}^g, \boldsymbol{\beta}^g\}$.

**Local Parameter Correction.** A local parameter correction step refines the SMPL parameters outputted by the global modeling step. The local correction step consists of a local transformer and a Hierarchical Spatial Correlation Regressor. The local transformer captures short-term local details in neighboring frames: nearby frames' static tokens are selected for short-term modeling, denoted as $\mathcal{S}^l = \{\boldsymbol{s}_t\}_{t=\frac{T}{2}-w}^{\frac{T}{2}+w}$ where $w$ represents the length of the selected neighborhood. When decoding, cross-attention is

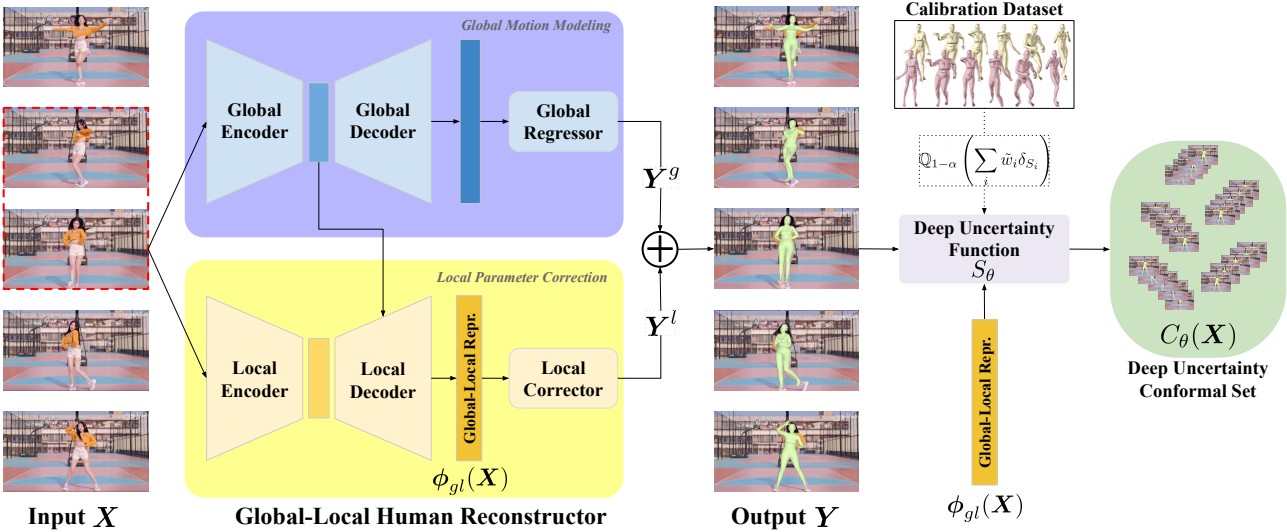

Figure 2: CUPS Overview. CUPS takes as input a sequence of input RGB video frames. The RGB video frames get encoded and fed into a global-local transformer human reconstruction model to produce SMPL parameters representing the human pose and shape in 3D as well as a decoupled global-local embedding. The output of the human reconstructor is supervised via SMPL loss. While training, we also learn a deep uncertainty function that learns to rank the uncertainty of the produced output sequence. Then after training, this deep uncertainty function is used as the conformity score for constructing a conformal set for conformal prediction.

applied to the query of the mid-frame token and key and value of the global encoder, capturing global human motion consistency and local, fine-grained human mesh structures.

**Global-Local Representation.** In the hierarchical spatial correlation regressor step, the model incorporates both the global prediction and a decoupled global-local representation into the regressor. The model learns joint correlations within the kinematic structure by modeling the local intra-frame human mesh structure, outputting a correction term $\boldsymbol{Y}^l = \{\boldsymbol{\theta}^l, \boldsymbol{\beta}^l\}$, where $\boldsymbol{\theta}^l$ is obtained using an MLP applied on the *decoupled global-local representation* $\boldsymbol{\phi}_{gl}(\boldsymbol{X})$ and global output $\boldsymbol{\theta}^g$, and $\boldsymbol{\beta}^l$ using $\boldsymbol{\phi}_{gl}(\boldsymbol{X})$ and $\boldsymbol{\beta}^g$. The human reconstruction model's final output SMPL values are obtained by adding the initial global prediction and the local correction output: $\boldsymbol{Y} := \{\boldsymbol{\theta}, \boldsymbol{\beta}\} = \boldsymbol{Y}^g + \boldsymbol{Y}^l$.

**Training Objective.** We follow previous works and apply standard L2 loss to the SMPL parameters and 3D/2D joints location (Kanazawa et al., 2018). We also follow the velocity loss on 3D/2D joint location proposed in (Shen et al., 2023) to learn consistency and capture the long-range dependency. We refer to the combined loss as $\mathcal{L}_G$.

### 4.2. Deep Uncertainty Function

Conformal predictors are calibrated using a nonconformity score function (Angelopoulos & Bates, 2021). Intuitively, this function measures to which extent a datapoint is unusual relative to a calibration dataset. The most common nonconformity functions are either simple residual terms

(Shafer & Vovk, 2008), raw logits (Stutz et al., 2021), or hand-designed functions (Yang & Pavone, 2023). A more recent line of work has demonstrated the benefits of using *learned nonconformity score function* (Zhang & Carlone, 2024), where the score is learned in an end-to-end manner with the machine learning model. In CUPS, we also learn the score end-to-end with the human reconstruction model. Moreover, as the nonconformity score measures how "unusual" the datapoint is relative to the calibration set, it provides an inherent uncertainty measure for the model: the lower the nonconformity score, the less uncertain the model is about the datapoint, and vice versa.

Formally, we wish to learn a function $S_\theta(\boldsymbol{X}, \boldsymbol{Y}) \in [0, 1]$ as the nonconformity score, which we refer to as the Deep Uncertainty Function.

**Definition 1** (Deep Uncertainty Function). *The Deep Uncertainty Function takes as input the decoupled global-local representation $\boldsymbol{\phi}_{gl}(\boldsymbol{X})$ as well as the corrected SMPL parameters $\boldsymbol{Y}$ and outputs a value between 0 and 1 using an MLP:*

$$S_\theta(\boldsymbol{X}, \boldsymbol{Y}) = \sigma\left(\boldsymbol{\phi}^{\mathrm{pred}}\right) \in [0, 1] \qquad (1)$$

*where $\boldsymbol{\phi}^{\mathrm{pred}} = \mathrm{MLP}(\boldsymbol{\phi}_{gl}(\boldsymbol{X}), \boldsymbol{\theta}, \boldsymbol{\beta})$ is the output from a multi-layer-perceptron and $\sigma$ is the sigmoid function.*

**Training Time Ensemble Augmentation.** To better learn the deep uncertainty function, we augment the model output with randomness such that the function learns to rank different samples. We achieve this by utilizing the intrinsic randomness in the human reconstruction model. Note

that the global static tokens $\mathcal{S}^g$ (and thus the global-local representation $\phi_{gl}(\boldsymbol{X})$) is obtained by randomly masking some portion of the input video frames. We augment each training step by randomly masking the video frames for $H_{\text{train}}$ times, effectively simulating $H_{\text{train}}$ hypotheses given the same input data. Thus, for each input video sequence $\boldsymbol{X}$, we get multiple hypotheses of SMPL parameters prediction: $\{\boldsymbol{Y}_i\}_{i=1}^{H_{\text{train}}}$ and global-local representation $\{\phi_{gl}(\boldsymbol{X})_i\}_{i=1}^{H_{\text{train}}}$. The proposed samples are used to train $S_\theta$.

**Training Objective.** The uncertainty function is implemented as a discriminator-style scoring function that measures the quality of the generated SMPL parameters conditioned on the input sequence, similar to the discriminator loss in (Kocabas et al., 2020). Thus, a *lower* score function output means the output is *more likely* (more realistic) to be from the ground-truth distribution in the embedding space. Formally, the deep uncertainty function optimizes the following loss:

$$\mathcal{L}_S = \mathbb{E}\left[S_\theta(\boldsymbol{X}, \boldsymbol{Y}_{\text{GT}})^2\right] + \mathbb{E}\left[(1 - S_\theta(\boldsymbol{X}, \boldsymbol{Y}))^2\right], \quad (2)$$

where $\boldsymbol{Y}_{\text{GT}}$ is the groundtruth SMPL parameters. Intuitively, this loss makes sure that samples close to the ground truth get higher conformity and vice versa, encouraging the scoring model to discriminate the prediction from the ground truth. Moreover, we also add an adversarial loss that will be back-propagated into the denoiser model, as done in (Kocabas et al., 2020; Zhang & Carlone, 2024; Goodfellow et al., 2020), which encourages the prediction model to output more realistic samples, adversarially confusing the discriminator:

$$\mathcal{L}_{\text{adv}} = \mathbb{E}\left[S_\theta(\boldsymbol{X}, \boldsymbol{Y})^2\right] \quad (3)$$

The overall training loss is a sum of SMPL and score loss: $\mathcal{L}_{\text{net}} = \mathcal{L}_G + \lambda(\mathcal{L}_S + \mathcal{L}_{\text{adv}})$, where $\lambda$ is a hyperparameter.

### 4.3. Conformal Human Reconstruction

Next we use the deep uncertainty function to quantify the uncertainty of the predicted human SMPL output. We first leverage the theoretical toolkit provided by (Barber et al., 2023), combining it with CUPS' deep uncertainty function, and then build on top of the coverage guarantee for nonexchangeable conformal prediction in (Barber et al., 2023) and provide two practical error bounds based on the characteristic of the video dataset and the design choice of the CUPS architecture.

**SMPL Conformal Calibration.** We introduce the calibration step post-training by using the score in Definition 1. For any prediction $\boldsymbol{Y}$ from the human reconstruction model and its corresponding input video sequence $\boldsymbol{X}$, $S_\theta(\boldsymbol{X}, \boldsymbol{Y})$ measures its "similarity" to an existing dataset — the nonconformity score for conformal calibration. When ex-

changeability holds (formally defined in Appendix A), the CP calibration is done by choosing a threshold using the $(1 - \alpha)$th quantile of the conformity scores calculated on the calibration set. Succinctly, we can define this threshold $\tau$ on the calibration dataset as follows:

$$\tau = \mathbb{Q}_{1-\alpha}\left(\sum_i \delta_{S_\theta(\boldsymbol{X}_i, \boldsymbol{Y}_i)}\right), \quad (4)$$

where $\delta_a$ represents the point mass at point $a$ and $\mathbb{Q}$ represents the quantile calculation. However, when the calibration is no longer exchangeable, such a threshold would not yield the desirable coverage guarantee of standard CP. Barber et al. (2023) propose to incorporate weighting terms in the CP calibration. Specifically, the new threshold $\tau^*$ is now calculated with prespecified weights $\tilde{w}$:

$$\tau^* = \mathbb{Q}_{1-\alpha}\left(\sum_i \tilde{w}_i \cdot \delta_{S_\theta(\boldsymbol{X}_i, \boldsymbol{Y}_i)}\right), \quad (5)$$

where $\tilde{w}_i \in [0, 1]$ denotes a prespecified weight placed on data point $i$. The values of $\tilde{w}_i$ is a design choice and it should intuitively be large for data with low nonconformity in the calibration. We discuss some practical design choices for the weights in the sections below.

**SMPL Conformal Prediction.** Once the threshold value $\tau^*$ is calibrated, we are able to do conformal prediction. Using $\tau^*$ defined in Equation (5) and the deep uncertainty function, for a datapoint $\boldsymbol{X}$, we define the Deep Uncertainty Conformal Set (DUCS) as the conformal prediction set.

**Definition 2** (DUCS). *The deep uncertainty conformal prediction set is the set of input-output pairs $X, Y$ such that the deep uncertainty value $S_\theta(\boldsymbol{X}, \boldsymbol{Y})$ is below the calibrated threshold value $\tau^*$*

$$C_\theta(\boldsymbol{X}) = \{\boldsymbol{Y} : S_\theta(\boldsymbol{X}, \boldsymbol{Y}) \leq \tau^*\}. \quad (6)$$

For a test video sequence $\boldsymbol{X}$ and predicted SMPL parameters $\boldsymbol{Y}$, it is straightforward to check its set membership in DUCS. More importantly, as we will show in Section 5.3, we can explicitly make a set prediction by using Monte Carlo Dropout during test time.

Under the framework of nonexchangeable conformal prediction (Barber et al., 2023), we analyze some theoretical properties of DUCS, which are amenable to our human reconstruction pipeline. We first define the tuple $\boldsymbol{Z}_i = (\boldsymbol{X}_i, \boldsymbol{Y}_i)$, which denotes the $i$-th example in our calibration dataset. We then construct the following sequence by combining the tuples $\boldsymbol{Z}_i$: $\boldsymbol{Z} = (\boldsymbol{Z}_1, \boldsymbol{Z}_2, \cdots, \boldsymbol{Z}_n)$ and define

$$\boldsymbol{Z}^i = (\boldsymbol{Z}_1, \cdots, \boldsymbol{Z}_{i-1}, \boldsymbol{Z}_n, \cdots, \boldsymbol{Z}_{n-1}, \boldsymbol{Z}_i), \quad (7)$$

which represents $\boldsymbol{Z}$ sequence after swapping the last point with the $i$-th calibration datapoint. Now, we define the weights $\tilde{w}_i$ needed for calibration (*cf.* (5)). As mentioned above, the exact weight formulation is a design choice, but to facilitate our analysis, we choose the following design based on the Euclidean distance of learned features.

**Definition 3** (Feature Distance Weight). *For our SMPL conformal calibration, the weight is defined based on the feature distance between the predicted SMPL feature and the ground-truth SMPL feature:*

$$w_i = \exp\left(-\frac{||\phi_i^{\mathrm{pred}} - \phi_i^{\mathrm{GT}}||^2}{\mathcal{T}}\right), \qquad (8)$$

*where $\mathcal{T}$ is the temperature hyperparameter, $\phi_i^{\mathrm{pred}} = \mathrm{MLP}(\boldsymbol{\phi}_{gl}(\boldsymbol{X}_i), \boldsymbol{\theta}_i, \boldsymbol{\beta}_i)$ is the predicted embedding in Definition 1 and $\phi_i^{\mathrm{GT}} = \mathrm{MLP}(\boldsymbol{\phi}_{gl}(\boldsymbol{X}_i), \boldsymbol{\theta}_i^{\mathrm{GT}}, \boldsymbol{\beta}_i^{\mathrm{GT}})$ is the ground-truth embedding.*

Then the quantile weights $\tilde{w}_i$ is the *normalized* version of $w_i$. The exact normalization technique is again a design choice that we describe below in Equation (11).

We now formally state the coverage guarantee of DUCS using Theorem 2 in (Barber et al., 2023). Without assuming exchangeability, DUCS is designed to be robust against distribution shifts.

**Theorem 1** (Nonexchangeable Coverage (Barber et al., 2023, Thm. 2)). *Under possibly non-exchangeable dataset distribution, the conformal prediction set defined in Definition 2 satisfies the following coverage guarantee:*

$$\mathbb{P}\left(\boldsymbol{Y} \in C_\theta(\boldsymbol{X})\right) \geq 1 - \alpha - \sum_{i=1}^n \tilde{w}_i \cdot D_{TV}\left(\boldsymbol{S}_\theta(\boldsymbol{Z}) \parallel \boldsymbol{S}_\theta(\boldsymbol{Z}^i)\right), \qquad (9)$$

*where $\tilde{w}_i$ is the normalized weight obtained via Definition 3, $D_{TV}(\cdot \parallel \cdot)$ represents the total variation distance, $\boldsymbol{S}_\theta(\boldsymbol{Z}) = [S_\theta(\boldsymbol{Z}_i)]_{i=1}^n$ and similarly for $\boldsymbol{S}_\theta(\boldsymbol{Z}^i)$.*

We provide a proof in Appendix B, following the outline in (Barber et al., 2023). The extra term $\sum_i^n \tilde{w}_i \cdot D_{\mathrm{TV}}\left(\boldsymbol{S}_\theta(\boldsymbol{Z}) \parallel \boldsymbol{S}_\theta(\boldsymbol{Z}^i)\right)$ is referred to as the *miscoverage gap*. Next, we present two practical bounds for the miscoverage gap by leveraging the structure of the datasets and architectures we use in our mesh estimation problems.

The first bound is borrowed from (Barber et al., 2023). We leverage the video dataset characteristics, assuming the distribution shift happens periodically. Weights are designed to account for this periodic change.

**Theorem 2** (Miscoverage under Periodic Change (Barber et al., 2023, 4.4)). *Using $w_i$ in Definition 3, we define the auxiliary weight $w_i'$:*

$$w_i' = \rho^{n+1-\pi(w_i)}, \qquad (10)$$

*where $\rho$ is a decay hyperparameter and $\pi(w_i)$ maps $w_i$ to its ranked position $\in [n]$ among all weights. Then the normalized weights are $\tilde{w}_i = \frac{w_i'}{\sum_j w_j'}$. Assuming that the most recent changepoint in the video dataset occurred $k$ time steps ago —such that $D_{TV}(\boldsymbol{Z}_i \parallel \boldsymbol{Z}_n) = 0$ for $i > n - k$ and could be arbitrarily large otherwise— we have the following bound:*

$$\sum_{i=1}^n \tilde{w}_i \cdot D_{TV}\left(\boldsymbol{S}_\theta(\boldsymbol{Z}) \parallel \boldsymbol{S}_\theta(\boldsymbol{Z}^i)\right) \leq \rho^k. \qquad (11)$$

This bound suggests that the miscoverage gap remains small as long as $k$ is large.

The second bound is novel and models the deep uncertainty output on the calibration set $\boldsymbol{S}_\theta(\boldsymbol{Z})$ using a beta distribution. In practice, this is achieved at test time using Monte Carlo Dropout (Gal & Ghahramani, 2016). The bound depends on the beta distributions formed by $\boldsymbol{S}_\theta(\boldsymbol{Z})$ and $\boldsymbol{S}_\theta(\boldsymbol{Z}^i)$.

**Theorem 3** (Miscoverage under Beta Distribution). *Assume the deep uncertainty values of the calibration set of size $n$ follow Beta distributions: $\boldsymbol{S}_\theta(\boldsymbol{Z}) \sim \beta(a_1, n - a_1)$, $\boldsymbol{S}_\theta(\boldsymbol{Z}^i) \sim \beta(a_2, n - a_2)$. If we assume that the difference between parameters $a_1$ and $a_2$ is bounded by $k$, we get the following bound without any assumption on the weights:*

$$\sum_{i=1}^n \tilde{w}_i \cdot D_{TV}\left(\boldsymbol{S}_\theta(\boldsymbol{Z}) \parallel \boldsymbol{S}_\theta(\boldsymbol{Z}^i)\right) \leq \sqrt{2 - 2\left(1 - \frac{2k}{n+k}\right)^{\frac{k}{2}}} \qquad (12)$$

This bound is stronger when $k$ is smaller. Intuitively, if the two distributions formed by swapping are similar then the miscoverage gap will be small. We refer the reader to Appendix C for the proof of Theorem 2 and 3.

## 5. Experiments

We provide a quantitative evaluation of our model against state-of-the-art baselines. We also provide ablation studies and qualitative results to support our design choices. We follow previous baselines (Shen et al., 2023; Kanazawa et al., 2018; 2019; Dwivedi et al., 2024; Choi et al., 2021) and report several intra-frame metrics, including Mean Per Joint Position Error (**MPJPE**), Procrustes-aligned MPJPE (**PA-MPJPE**), and Mean Per Vertex Position Error (**MPVPE**). Following (Shen et al., 2023; Choi et al., 2021), we also provide a result for the second-order acceleration error (**Accel**) for the inter-frame smoothness.

### 5.1. Baselines Comparisons

We follow the same dataset split and setup as done in previous works and evaluated on 3DPW (Von Marcard

| Method | 3DPW (Von Marcard et al., 2018) | | | | MPI-INF-3DHP (Wei et al., 2022) | | | Human3.6M (Ionescu et al., 2013) | | | # Frames |
|---|---|---|---|---|---|---|---|---|---|---|---|
| | PA-MPJPE ↓ | MPJPE ↓ | MPVPE ↓ | Accel ↓ | PA-MPJPE ↓ | MPJPE ↓ | Accel ↓ | PA-MPJPE ↓ | MPJPE ↓ | Accel ↓ | |
| VIBE (Kocabas et al., 2020) | 57.6 | 91.9 | - | 25.4 | 68.9 | 103.9 | 27.3 | 53.3 | 78.0 | 27.3 | 16 |
| MEVA (Luo et al., 2020) | 54.7 | 86.9 | - | 11.6 | 65.4 | 96.4 | 11.1 | 53.2 | 76.0 | 15.3 | 90 |
| TCMR (Choi et al., 2021) | 52.7 | 86.5 | 102.9 | 7.1 | 63.5 | 97.3 | 8.5 | 52.0 | 73.6 | 3.9 | 16 |
| MPS-Net (Wei et al., 2022) | 52.1 | 84.3 | 99.7 | 7.4 | 62.8 | 96.7 | 9.6 | 47.4 | 69.4 | 3.6 | 16 |
| POCO (Dwivedi et al., 2024) | 50.5 | 80.5 | 96.5 | 6.7 | 62.1 | 93.7 | 8.1 | 46.4 | 68.1 | 3.6 | 1 |
| GLoT (Shen et al., 2023) | 50.6 | 80.7 | 96.3 | **6.6** | 61.5 | 93.9 | 7.9 | 46.3 | 67.0 | 3.6 | 16 |
| CUPS (Ours) | **48.7** | **76.2** | **91.7** | 6.9 | **61.3** | **92.8** | **7.2** | **44.0** | **63.8** | **3.5** | 16 |

Table 1: Multiple errors (↓) results on 3DPW, MPI-INF-3DHP, and Human3.6M. All methods use 3DPW training set for training. Comparisons show that CUPS outperforms other baseline methods in the vast majority of metrics.

et al., 2018), Human3.6M (Ionescu et al., 2013), and MPII-3DHP (Mehta et al., 2017). More details on the construction of training dataset are in Appendix D. As shown in Table 1, our model outperforms state-of-the-art baseline methods. On 3DPW, for example, we outperform GLoT's PA-MPJPE by 1.9mm, MPJPE by 4.5mm, and MPVPE by 4.6mm. While Accel performance was slightly worse off on 3DPW, on the other two datasets, our method surpasses baselines on all metrics. Moreover, our method outperforms GLoT by a noticeable margin, indicating that the deep uncertainty function $S_\theta$ is important in that it forces the human reconstruction to output higher-quality samples during training. In Table 2, following previous works (Choi et al., 2021; Dwivedi et al., 2024; Shen et al., 2023), we measure our method's generalizability to unseen datasets. In this set of experiments, none of the methods uses 3DPW dataset during training. Again, our method outperforms baselines by a noticeable margin.

| Method | 3DPW | | | |
|---|---|---|---|---|
| | PA-MPJPE ↓ | MPJPE ↓ | MPVPE ↓ | Accel ↓ |
| HMR (Kanazawa et al., 2018) | 76.7 | 130.0 | - | 37.4 |
| 3DMB (Biggs et al., 2020) | 74.9 | 120.8 | - | - |
| SPIN (Kolotouros et al., 2019) | 59.2 | 96.9 | 116.4 | 29.8 |
| HMMR (Kanazawa et al., 2019) | 72.6 | 116.5 | 139.3 | 15.2 |
| VIBE (Kocabas et al., 2020) | 56.5 | 93.5 | 113.4 | 27.1 |
| TCMR (Choi et al., 2021) | 55.8 | 95.0 | 111.5 | 7.0 |
| MPS-Net (Wei et al., 2022) | 54.0 | 91.6 | 109.6 | 7.5 |
| POCO (Dwivedi et al., 2024) | 54.7 | 89.3 | 108.4 | 6.8 |
| GLoT (Shen et al., 2023) | 53.5 | 89.9 | 107.8 | 6.7 |
| CUPS (Ours) | **53.0** | **85.7** | **103.6** | **6.6** |

Table 2: Multiple errors (↓) results on 3DPW. None of the methods use 3DPW for training. CUPS outperforms all baselines.

### 5.2. Ablation Studies

**Training Time Ensemble Augmentation.** We augment the training dataset online to better train the deep uncertainty function by leveraging the intrinsic stochasticity (*e.g.*, frame masking) of the human reconstruction model. We compare results as a function of the number of samples ($H$). When $H = 1$, there is no augmentation and we are just running the forward pass once. We train several models using a range of $H$ values and evaluate the MPJPE using these models in Figure 3. Results suggest that using more proposed samples during training reduces test error overall and the improvement saturates after 30.

**Choice of Conformity Score Function.** We compare our proposed deep uncertainty function (**DUF**) trained using adversarial loss with several different losses in Figure 4: a score function augmented with inefficiency loss (**Ineff.**) (Stutz et al., 2021) during training and a classifier-style conformity score function (**Class.**). From Figure 4, we see that DUF and inefficiency-augmented DUF result in similar performance quantitatively. While the classifier-style loss performs better than without any scoring function quantitatively, the predicted mesh shape on the videos is less realistic. Please refer to Appendix E for more details.

**Strength of the Uncertainty Loss.** Finally, we ablate the hyperparameter of the training loss for $S_\theta(\boldsymbol{X}, \boldsymbol{Y})$, $\lambda$, in the overall training objective $\mathcal{L}_{\text{net}}$. This is an important ablation in that we can find a suitable scale of the loss for the deep uncertainty function to make sure it does not conflict with the pose loss optimization. Results in Figure 4 suggest that 0.6 is the most efficient strength across all values, as a smaller scale does not train the scoring model sufficiently and a higher scale conflicts with the pose loss.

### 5.3. Monte Carlo Dropout

One interesting byproduct of learning the deep uncertainty function $S_\theta(\boldsymbol{X}, \boldsymbol{Y})$ is that we can construct the DUCS $C_\theta(\boldsymbol{X})$ explicitly by sampling the output SMPL parameters multiple times, just like during training time ensemble augmentation. While the model itself is not exactly probabilistic, we can emulate its stochasticity during inference time with Monte Carlo Dropout which lends itself to modeling the model uncertainty in a Bayesian way (Gal & Ghahramani, 2016). This procedure effectively enables us to make multi-hypothesis predictions during test time, and by checking the set membership of each hypothesis, we are able to explicitly construct the DUCS $C_\theta(\boldsymbol{X})$ with minimal changes to the model. Prediction sets from using MC Dropout are shown in Figure 6

### 5.4. In-the-Wild Videos

To test the generalizability of our method to in-the-wild videos, we collect videos from YouTube and TikTok. We directly apply the CUPS model trained on the 3DPW

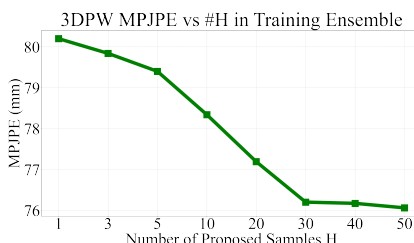

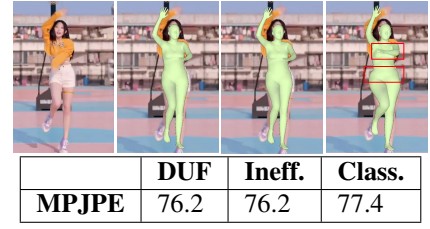

|  | DUF | Ineff. | Class. |
|---|---|---|---|
| **MPJPE** | 76.2 | 76.2 | 77.4 |

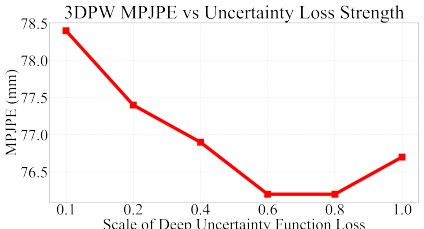

Figure 3: Comparison of nr. of samples proposed during training time ensemble.

Figure 4: Conformity scores choices on 3DPW (bottom) and internet videos (top).

Figure 5: Comparison of strength of uncertainty loss in the total training loss.

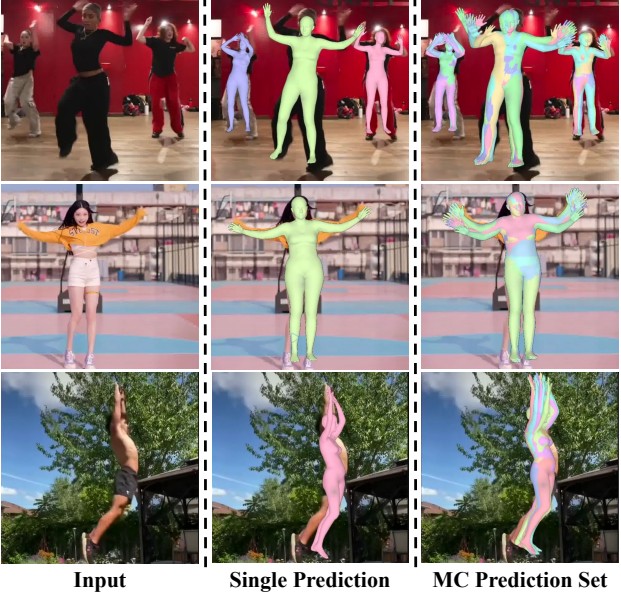

| **Input** | **Single Prediction** | **MC Prediction Set** |

Figure 6: In the wild video SMPL predictions with both single hypothesis and multiple hypotheses using MC Dropout.

dataset to test on in-the-wild videos. We run both regular CUPS and CUPS with MC Dropout for multiple hypotheses results are shown in Figure 6, where the input videos are collected from TikTok. For 3D visualization, please refer to **this anonymized website** to interact with CUPS predictions in 3D.

### 5.5. Empirical Coverage

Here we test the empirical coverage of the deep uncertainty function using the three testing datasets in Table 1. Mathematically, we calculate the following value:

$$\bar{C} = \frac{1}{|I_{\text{test}}|} \sum_{\boldsymbol{Y}^{\text{GT}} \sim I_{\text{test}}} \mathbb{1}\left(\boldsymbol{Y}^{\text{GT}} \in C_\theta(\boldsymbol{X})\right) \qquad (13)$$

and compare against the desired coverage value 1-$\alpha$. We use $\alpha = 0.1$ for CP, calibrating with the 90th quantile. Furthermore, we compare the performance with weighted calibration using weights defined in Definition 3 against unweighted calibration as done in regular CP. From Table 3, we see the empirical coverage for weighted CP is around

$88\% \pm 3\%$ for all three datasets, which remains close to $1 - \alpha$, and in some cases, it exceeds this value. Weighted CP coverage is noticeably higher than unweighted CP, corroborating the results in (Barber et al., 2023). The coverage result with weighted CP is encouraging because it illustrates that the miscoverage gap is small in all three datasets.

|  | **3DPW** | **3DHP** | **H3.6M** |
|---|---|---|---|
| **Weighted CP** | $86.2 \pm 2.1\%$ | $87.3 \pm 2.2\%$ | $89.0 \pm 1.5\%$ |
| **Theorem 2 Bound** | $\geq 83.9\%$ | $\geq 84.9\%$ | $\geq 85.8\%$ |
| **Theorem 3 Bound** | $\geq 84.0\%$ | $\geq 85.3\%$ | $\geq 86.8\%$ |
| **Regular CP** | $81.0 \pm 3.4\%$ | $83.2 \pm 2.8\%$ | $85.2 \pm 2.3\%$ |

Table 3: Empirical coverage with weighted vs. regular calibration using the learned deep uncertainty function on three different datasets. Bounds are obtained as described in Appendix B.

### 5.6. Implementation and Training Details

The reconstruction model takes as input video sequences of length 16, following (Shen et al., 2023). We use an Adam optimizer with a weight decay of 0.1 and a momentum of 0.9. The adversarial loss weight is 0.6 and is optimized every 100 iterations. Our model is trained using an NVIDIA V100 GPU, where training consumes an amortized GPU memory of 20GB, and CPU memory of 160 GB. We train the model for 100 epochs with an initial learning rate of 5e-5 with a cosine scheduler. The ensemble augmentation step produces 20 samples for the same input datapoint.

## 6. Conclusion

We presented CUPS, an approach for learning sequence-to-sequence 3D human shapes and poses from RGB videos with uncertainty quantification. Our method uses a deep uncertainty function that is trained end-to-end with the 3D pose-shape estimator. The deep uncertainty function computes a conformity score, enabling the calibration of a conformal predictor to assess the quality of output predictions at inference time. We present two practical bounds for the miscoverage gap in CP, providing theoretical backing for the uncertainty quantification of our model. Our results demonstrate that CUPS achieves state-of-the-art performance across various metrics and datasets, while inheriting the probabilistic guarantees of conformal prediction.

## Acknowledgements

This work was partially funded by Amazon Robotics under the "Safe Autonomy Leveraging Intelligent Optimization" program, by the ARL DCIST program, and by the ONR RAPID program.

## Impact Statement

This paper presents a deep-learning system on improving human mesh recovery by utilizing deep uncertainty function and theoretical guarantees. It can help quantify the uncertainty of deep learning models when adopted in practice. In addition, our framework is fully modular, and it aims to advance research of architecture innovation and promote applications of UQ.

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

# Supplementary Material

## A. Exchangeable Distributions

First, we define exchangeability in a probabilistic distribution:

**Definition 4** (Exchangeability in Probabilistic Distribution). *A sequence of random variables $X_1, X_2, \ldots, X_n$ is said to have an exchangeable distribution if the joint distribution of $X_1, X_2, \ldots, X_n$ is invariant under any permutation of indices. Formally, for any permutation $\pi$ of $\{1, 2, \ldots, n\}$,*

$$P(X_1 = x_1, X_2 = x_2, \ldots, X_n = x_n) = P(X_{\pi(1)} = x_1, X_{\pi(2)} = x_2, \ldots, X_{\pi(n)} = x_n),$$

*for all possible values $x_1, x_2, \ldots, x_n$ of $X_1, X_2, \ldots, X_n$.*

## B. Proof of Coverage Guarantee without Exchangeability

We first recall the following definition of Deep Uncertainty Conformal Set:

**Definition 2** (DUCS). *The deep uncertainty conformal prediction set is the set of input-output pairs $X, Y$ such that the deep uncertainty value $S_\theta(X, Y)$ is below the calibrated threshold value $\tau^*$*

$$C_\theta(X) = \{Y : S_\theta(X, Y) \leq \tau^*\}. \tag{6}$$

We further define tuple $Z_i = (X_i, Y_i)$ and

$$Z = (Z_1, \cdots, Z_{n+1}),$$

as well as

$$Z^i = (Z_1, \cdots, Z_{i-1}, Z_{n+1}, \cdots, Z_n, Z_i),$$

which represents $Z$ sequence after swapping the test point with the $i$-th calibration point.

**Lemma 4** (Weight sum upper bound (Harrison, 2012, Lemma 3)). *For all $\alpha, w_1, \cdots, w_n \in [0, \infty]$, and all $t_1, \cdots, t_{n+1} \in [-\infty, \infty]$, we have:*

$$\sum_{k=1}^{n+1} w_k \mathbf{1} \left( \sum_{i=1}^{n+1} w_i \mathbf{1}(t_i \geq t_k) \leq \alpha \right) \leq \alpha$$

*Proof.* We follow the sketch proof in (Harrison, 2012) and provide detailed proof for interested readers. The tuples $(t_i, w_i)$ can be permuted without affecting the value of the LHS, so we can assume that $t_1 \geq \cdots \geq t_n$. This implies that $\sum_{i=1}^{n+1} w_i \mathbf{1}(t_i \geq t_k)$ is increasing in $k$. There exists a $k^*$ defined as follows:

$$k^* = \sup_k \sum_{i=1}^{n+1} w_i \mathbf{1}(t_i \geq t_k) \leq \alpha$$

Thus, we have:

$$\sum_{k=1}^{n+1} w_k \mathbf{1} \left( \sum_{i=1}^{n+1} w_i \mathbf{1}(t_i \geq t_k) \leq \alpha \right) = \sum_{k=1}^{k^*} w_k = \sum_{i=1}^{k^*} w_i \mathbf{1}(t_i \geq t_{k^*}) \leq \alpha \tag{14}$$

Next, recall the nonexchangeable conformal prediction coverage guarantee:

**Theorem 1** (Nonexchangeable Coverage (Barber et al., 2023, Thm. 2)). *Under possibly non-exchangeable dataset distribution, the conformal prediction set defined in Definition 2 satisfies the following coverage guarantee:*

$$\mathbb{P}(Y \in C_\theta(X)) \geq 1 - \alpha - \sum_{i=1}^{n} \tilde{w}_i \cdot D_{TV}\left(S_\theta(Z) \| S_\theta(Z^i)\right), \tag{9}$$

*where $\tilde{w}_i$ is the normalized weight obtained via Definition 3, $D_{TV}(\cdot \| \cdot)$ represents the total variation distance, $S_\theta(Z) = [S_\theta(Z_i)]_{i=1}^n$ and similarly for $S_\theta(Z^i)$.*

*Proof.* We follow the sketch proof in (Barber et al., 2023) and provide detailed proof for interested readers. We look at the complement of the above probability $\mathbb{P}\left(\boldsymbol{Y}_{n+1} \in C_\theta(\boldsymbol{X}_{n+1})\right)$. For simplicity, we first define $S_i := S_\theta(\boldsymbol{X}_i, \boldsymbol{Y}_i)$. By definition of DUCS, we have:

$$
\begin{aligned}
\boldsymbol{Y}_{n+1} \notin C_\theta(\boldsymbol{X}_{n+1}) &\Leftrightarrow S_{n+1} > \mathbb{Q}_{1-\alpha}\left(\sum_{i=1}^{n+1} \tilde{w}_i \cdot \delta_{S_i}\right) \\
&\Leftrightarrow \mathbb{Q}_{1-\alpha}\left(\sum_{i=1}^{n} \tilde{w}_i \cdot \delta_{S_i} + \tilde{w}_{n+1} \cdot \delta_{+\infty}\right)
\end{aligned}
\tag{15}
$$

Then define an *unusual set* function $\mathcal{U}$:

$$
\mathcal{U}(S) = \left\{i \in [n+1] : S_i > \mathbb{Q}_{1-\alpha}\left(\sum_{i=1}^{n+1} \tilde{w}_i \cdot \delta_{S_i}\right)\right\},
\tag{16}
$$

which represents the indices $i$ where the deep nonconformity score values are too large. Then we know that noncoverage of $\boldsymbol{Y}_{n+1}$ implies the unusualness of point $k$:

$$
\boldsymbol{Y}_{n+1} \notin C_\theta(\boldsymbol{X}_{n+1}) \rightarrow k \in \mathcal{U}(\boldsymbol{S}_\theta(\boldsymbol{Z}^k)).
\tag{17}
$$

Thus, we have:

$$
\begin{aligned}
\mathbb{P}(k \in \mathcal{U}(\boldsymbol{S}_\theta(\boldsymbol{Z}^k))) &= \sum_{i=1}^{n+1} \mathbb{P}\left(k = i, i \in \mathcal{U}(\boldsymbol{S}_\theta(\boldsymbol{Z}^i))\right) \\
&= \sum_{i=1}^{n+1} \tilde{w}_i \cdot \mathbb{P}(i \in \mathcal{U}(\boldsymbol{S}_\theta(\boldsymbol{Z}^i))) \\
&\leq \sum_{i=1}^{n+1} \tilde{w}_i \left(\mathbb{P}(i \in \mathcal{U}(\boldsymbol{S}_\theta(\boldsymbol{Z})) + D_{\mathrm{TV}}\left(\boldsymbol{S}_\theta(\boldsymbol{Z}) \| \boldsymbol{S}_\theta(\boldsymbol{Z}^i)\right)\right) \\
&= \mathbb{E}\left[\sum_{i \sim \mathcal{U}(\boldsymbol{S}_\theta(\boldsymbol{Z}^i))} \tilde{w}_i\right] + \sum_{i=1}^{n} \tilde{w}_i D_{\mathrm{TV}}\left(\boldsymbol{S}_\theta(\boldsymbol{Z}) \| \boldsymbol{S}_\theta(\boldsymbol{Z}^i)\right) \\
&\leq \alpha + \sum_{i=1}^{n} \tilde{w}_i D_{\mathrm{TV}}\left(\boldsymbol{S}_\theta(\boldsymbol{Z}) \| \boldsymbol{S}_\theta(\boldsymbol{Z}^i)\right) \quad \text{[by Lemma 4]}
\end{aligned}
\tag{18}
$$

By complement, $\mathbb{P}\left(\boldsymbol{Y}_{n+1} \in C_\theta(\boldsymbol{X}_{n+1})\right) \geq 1 - \alpha - \sum_{i=1}^{n} \tilde{w}_i \cdot D_{\mathrm{TV}}\left(\boldsymbol{S}_\theta(\boldsymbol{Z}) \| \boldsymbol{S}_\theta(\boldsymbol{Z}^i)\right)$

## C. Upper Bounds of Coverage Gap

We provide two possible bounds for the coverage gap. The first bound comes from an example usage in Barber et al. (2023) and the second bound is derived by assuming the outputs follow a beta distribution.

### C.1. Bounded Periodic Changes

We are interested in finding an upper bound of the coverage gap in Theorem 1. Specifically, we are trying to bound $D_{\mathrm{TV}}\left(\boldsymbol{S}_\theta(\boldsymbol{Z}) \| \boldsymbol{S}_\theta(\boldsymbol{Z}^i)\right)$. One interesting case to analyze for video data is that we might have periodic large changes in the distribution rather than a gradual drift (i.e. there might be a changepoint in the calibration set).

**Assumption 1** ($k$-step changepoints in dataset). *Suppose that the most recent changepoint occurred $k$ steps ago, so that $D_{TV}(\boldsymbol{S}_\theta(\boldsymbol{Z}_{n+1}) \| \boldsymbol{S}_\theta(\boldsymbol{Z}_i)) \rightarrow 0$ for $i > n - k$. However, we might have an arbitrarily large difference from the new test point before the changepoint step: $D_{TV}(\boldsymbol{S}_\theta(\boldsymbol{Z}_{n+1}) \| \boldsymbol{S}_\theta(\boldsymbol{Z}_i)) \rightarrow 1$ for $i \leq n - k$.*

Now, we try to bound the coverage gap from Theorem 1. We first design the weights amenable to our analysis. Recall the definition of weights:

**Definition 3** (Feature Distance Weight). *For our SMPL conformal calibration, the weight is defined based on the feature distance between the predicted SMPL feature and the ground-truth SMPL feature:*

$$w_i = \exp\left(-\frac{||\phi_i^{\mathrm{pred}} - \phi_i^{\mathrm{GT}}||^2}{\mathcal{T}}\right), \tag{8}$$

*where $\mathcal{T}$ is the temperature hyperparameter, $\phi_i^{\mathrm{pred}} = \mathrm{MLP}(\phi_{gl}(\boldsymbol{X}_i), \boldsymbol{\theta}_i, \boldsymbol{\beta}_i)$ is the predicted embedding in Definition 1 and $\phi_i^{\mathrm{GT}} = \mathrm{MLP}(\phi_{gl}(\boldsymbol{X}_i), \boldsymbol{\theta}_i^{\mathrm{GT}}, \boldsymbol{\beta}_i^{\mathrm{GT}})$ is the ground-truth embedding.*

Again, recall the miscoverage gap under bounded period changes:

**Theorem 2** (Miscoverage under Periodic Change (Barber et al., 2023, 4.4)). *Using $w_i$ in Definition 3, we define the auxiliary weight $w_i'$:*

$$w_i' = \rho^{n+1-\pi(w_i)}, \tag{10}$$

*where $\rho$ is a decay hyperparameter and $\pi(w_i)$ maps $w_i$ to its ranked position $\in [n]$ among all weights. Then the normalized weights are $\tilde{w}_i = \frac{w_i'}{\sum_j w_j'}$. Assuming that the most recent changepoint in the video dataset occurred $k$ time steps ago —such that $D_{TV}(\boldsymbol{Z}_i \parallel \boldsymbol{Z}_n) = 0$ for $i > n - k$ and could be arbitrarily large otherwise— we have the following bound:*

$$\sum_{i=1}^{n} \tilde{w}_i \cdot D_{TV}\left(\boldsymbol{S}_\theta(\boldsymbol{Z}) \parallel \boldsymbol{S}_\theta(\boldsymbol{Z}^i)\right) \leq \rho^k. \tag{11}$$

*Proof.* We follow the sketch proof in (Barber et al., 2023) and provide detailed proof for interested readers. From Assumption 1, suppose the variation before the changepoint could be arbitrarily large:

$$
\begin{aligned}
\sum_{i=1}^{n} \tilde{w}_i D_{\mathrm{TV}}\left(\boldsymbol{S}_\theta(\boldsymbol{Z}) \parallel \boldsymbol{S}_\theta(\boldsymbol{Z}^i)\right) &\leq \sum_{i=1}^{n-k} \tilde{w}_i \\
&= \frac{\sum_{i=1}^{n-k} \rho^{n+1-\pi(i)}}{1 + \sum_{i=1}^{n} \rho^{n+1-\pi(i)}} \\
&\leq \rho^k \cdot \frac{1 - \rho^{n-k}}{1 - \rho^n} \\
&\leq \rho^k
\end{aligned} \tag{19}
$$

Intuitively, this tells us that the coverage gap will be small as long as $k$ is large - namely, as long as we have enough data after the changepoint.

**Remark 1** (Measuring $\rho$ and $k$). *As explained in Barber et al. (2023), $\rho$ is a decay parameter less than 1 and the above miscoverage gap is small as long as $k$ is sufficiently large. To measure $k$, for each of the three test datasets in Table 3, we measure the average number of video sequences between the two closest sequence datapoints belonging to different subjects/activities.*

### C.2. Bounded Covariates

Next, we are trying to bound $D_{\mathrm{TV}}\left(\boldsymbol{S}_\theta(\boldsymbol{Z}) \parallel \boldsymbol{S}_\theta(\boldsymbol{Z}^i)\right)$ under distributional modeling. To find this bound, we make use of the **Hellinger distance**. The Hellinger distance $H^2(\boldsymbol{P}, \boldsymbol{Q})$ between two probability measure $\boldsymbol{P}$ and $\boldsymbol{Q}$ on a measure space $\mathcal{X}$ with respect to an auxiliary measure $\lambda$ (e.g. joint) is defined as:

$$H^2(\boldsymbol{P}, \boldsymbol{Q}) = \frac{1}{2} \int_{\mathcal{X}} \left(\sqrt{p(x)} - \sqrt{q(x)}\right)^2 \lambda(dx),$$

where $\boldsymbol{P}(dx) = p(x)\lambda(dx)$ and $\boldsymbol{Q}(dx) = q(x)\lambda(dx)$. Succinctly, we can denote:

$$H(\boldsymbol{P}, \boldsymbol{Q}) = \frac{1}{\sqrt{2}} ||\sqrt{\boldsymbol{P}} - \sqrt{\boldsymbol{Q}}||_2$$

It turns out that we can make use of the Hellinger distance to bound the total variation distance.

$$D_{\text{TV}}\left(\boldsymbol{P} \parallel \boldsymbol{Q}\right) \leq \sqrt{2}H(\boldsymbol{P}, \boldsymbol{Q}) \tag{20}$$

*Proof.* This is a fairly well-known results in statistics and we provide a detailed proof for the sake of completeness.

$$\begin{aligned}
D_{\text{TV}}^2\left(\boldsymbol{P} \parallel \boldsymbol{Q}\right) &= \frac{1}{4}\left(\sum_i |p_i - q_i|^2\right) \\
&= \frac{1}{4}\left(\sum_i (\sqrt{p_i} - \sqrt{q_i})(\sqrt{p_i} + \sqrt{q_i})\right)^2 \\
&\leq \frac{1}{4}\left(\sum_i (\sqrt{p_i} - \sqrt{q_i})^2\right)\left(\sum_i (\sqrt{p_i} + \sqrt{q_i})^2\right) \\
&\leq \frac{1}{2}H^2(\boldsymbol{P}, \boldsymbol{Q})\left(2 + 2\sum_i \sqrt{p_i}\sqrt{q_i}\right) \\
&\leq H^2(\boldsymbol{P}, \boldsymbol{Q})(2 - H^2(\boldsymbol{P}, \boldsymbol{Q})) \leq 2H^2(\boldsymbol{P}, \boldsymbol{Q})
\end{aligned} \tag{21}$$

Hence, $D_{\text{TV}}\left(\boldsymbol{P} \parallel \boldsymbol{Q}\right) \leq \sqrt{2}H(\boldsymbol{P}, \boldsymbol{Q})$

We can now focus on providing an upper-bound Hellinger distance instead. Since our conformity score is outputted by a sigmoid function via Monte Carlo Dropout during test time, one reasonable assumption is that the conformity score outputs follow a Beta distribution.

**Assumption 2** (Conformity scores follow a Beta distribution).

$$\boldsymbol{S}_\theta(\boldsymbol{Z}) \sim \beta(a_1, n - a_1), \quad \boldsymbol{S}_\theta(\boldsymbol{Z}^i) \sim \beta(a_2, n - a_2)$$

*where $a_1$ and $a_2$ are defined by the permuted dataset. Without loss of generality, we then assume:*

$$a_1 - k \leq a_2 \leq a_1 + k$$

*That is, assume the difference in the Beta parameters is bounded.*

The assumption makes intuitive sense in that it can be thought of as the proportion of calibration data points that are close to (*conform to*) the new test point. Furthermore, the expected proportion change between the original and permuted dataset is bounded by $\frac{k}{n}$. This assumption makes sense in that we are essentially assuming that after swapping one pair of data points, the change in proportion of data that conform to the test data is bounded.

With the Beta distributions defined, recall the bound to prove:

**Theorem 3** (Miscoverage under Beta Distribution). *Assume the deep uncertainty values of the calibration set of size $n$ follow Beta distributions: $\boldsymbol{S}_\theta(\boldsymbol{Z}) \sim \beta(a_1, n - a_1), \quad \boldsymbol{S}_\theta(\boldsymbol{Z}^i) \sim \beta(a_2, n - a_2)$. If we assume that the difference between parameters $a_1$ and $a_2$ is bounded by $k$, we get the following bound without any assumption on the weights:*

$$\sum_{i=1}^n \tilde{w}_i \cdot D_{TV}\left(\boldsymbol{S}_\theta(\boldsymbol{Z}) \parallel \boldsymbol{S}_\theta(\boldsymbol{Z}^i)\right) \leq \sqrt{2 - 2\left(1 - \frac{2k}{n+k}\right)^{\frac{k}{2}}} \tag{12}$$

*Proof.* First, we can express the Hellinger distance between two Beta-distributed measures in closed form:

$$H^2(\boldsymbol{S}_\theta(\boldsymbol{Z}), \boldsymbol{S}_\theta(\boldsymbol{Z}^i)) = 1 - \frac{B\left(\frac{a_1+a_2}{2}, n - \frac{a_1+a_2}{2}\right)}{\sqrt{B(a_1, n - a_1)B(a_2, n - a_2)}}, \tag{22}$$

where $B(m, n)$ is the **beta function** defined as:

$$\begin{aligned}
B(m, n) &= \frac{\Gamma(m)\Gamma(n)}{\Gamma(m + n)} \\
&= \frac{(m - 1)!(n - 1)!}{(m + n - 1)!} = \frac{\frac{m+n}{mn}}{\binom{m+n}{n}}
\end{aligned} \tag{23}$$

We define a short-hand notation $a := \frac{a_1+a_2}{2}$, and then by the definition of beta function, the numerator of Eq. 22 becomes:

$$B\left(\frac{a_1+a_2}{2}, n - \frac{a_1+a_2}{2}\right) = \frac{(a-1)!(n-1-a)!}{(n-1)!} \tag{24}$$

Similarly, in the denominator:

$$B(a_1, n-a_1) = \frac{(a_1-1)!(n-1-a_1)!}{(n-1)!}, \quad B(a_2, n-a_2) = \frac{(a_2-1)!(n-1-a_2)!}{(n-1)!} \tag{25}$$

Combining the expressions above and plugging them into Eq. 22, we have:

$$
\begin{aligned}
H^2(\boldsymbol{S}_\theta(\boldsymbol{Z}), \boldsymbol{S}_\theta(\boldsymbol{Z}^i)) &= 1 - \frac{B\left(\frac{a_1+a_2}{2}, n-\frac{a_1+a_2}{2}\right)}{\sqrt{B(a_1, n-a_1)B(a_2, n-a_2)}} \\
&= 1 - \frac{\frac{(a-1)!(n-1-a)!}{(n-1)!}}{\sqrt{\frac{(a_1-1)!(n-1-a_1)!}{(n-1)!} \cdot \frac{(a_2-1)!(n-1-a_2)!}{(n-1)!}}} \\
&= 1 - \frac{(a-1)!(n-1-a)!}{\sqrt{(a_1-1)!(a_2-1)!(n-1-a_1)!(n-1-a_2)!}}
\end{aligned} \tag{26}
$$

With Assumption 2 in place, we are able to bound the fraction in Eq. 26 as follows:

$$
\begin{aligned}
\left(\frac{(a-1)!(n-1-a)!}{\sqrt{(a_1-1)!(a_2-1)!(n-1-a_1)!(n-1-a_2)!}}\right)^2 &= \frac{\Gamma(a)\Gamma(a)\Gamma(n-a)\Gamma(n-a)}{\Gamma(a_1)\Gamma(a_2)\Gamma(n-a_1)\Gamma(n-a_2)} \\
\text{[by bounded difference between } a_1 \text{ and } a_2] \quad &\geq \frac{\Gamma(a_1+\frac{k}{2})\Gamma(a_1+\frac{k}{2})\Gamma(n-a_1-\frac{k}{2})\Gamma(n-a_1-\frac{k}{2})}{\Gamma(a_1)\Gamma(a_1+k)\Gamma(n-a_1)\Gamma(n-a_1-k)} \\
\text{[by definition of } \Gamma \text{ function]} \quad &\geq a_1^{\frac{k}{2}} \cdot (a_1+k)^{-\frac{k}{2}} \cdot (n-a_1)^{-\frac{k}{2}} \cdot (n-a_1-k)^{\frac{k}{2}} \\
\text{[rearranging the terms]} \quad &\geq \left(\frac{a_1 \cdot (n-a_1-k)}{(a_1+k)\cdot(n-a_1)}\right)^{\frac{k}{2}} \\
\text{[expanding the terms]} \quad &\geq \left(\frac{a_1 n - a_1^2 - a_1 k}{a_1 n - a_1^2 - a_1 k + kn}\right)^{\frac{k}{2}} \\
&\geq \left(1 - \frac{kn}{a_1 n - a_1^2 - a_1 k + kn}\right)^{\frac{k}{2}} \\
\text{[denominator extremum at } a_1 = \frac{n-k}{2}] \quad &\geq \left(1 - \frac{kn}{\frac{(n-k)^2}{4} + kn}\right)^{\frac{k}{2}} \\
&\geq \left(\frac{(n-k)^2}{(n-k)^2 + 4kn}\right)^{\frac{k}{2}} \\
&\geq \left(\frac{n-k}{n+k}\right)^k \\
\text{Hence, } \frac{(a-1)!(n-1-a)!}{\sqrt{(a_1-1)!(a_2-1)!(n-1-a_1)!(n-1-a_2)!}} &\geq \left(1 - \frac{2k}{n+k}\right)^{\frac{k}{2}}
\end{aligned} \tag{27}
$$

Putting it all together, we know that

$$1 - H^2(\boldsymbol{S}_\theta(\boldsymbol{Z}), \boldsymbol{S}_\theta(\boldsymbol{Z}^i)) \geq \left(1 - \frac{2k}{n+k}\right)^{\frac{k}{2}}$$

and equivalently,

$$H^2(\boldsymbol{S}_\theta(\boldsymbol{Z}), \boldsymbol{S}_\theta(\boldsymbol{Z}^i)) \leq 1 - \left(1 - \frac{2k}{n+k}\right)^{\frac{k}{2}}$$

Hence, we have the final upper bound on the coverage gap via Lemma 20:

$$D_{\mathrm{TV}}\left(\boldsymbol{S}_\theta(\boldsymbol{Z}) \parallel \boldsymbol{S}_\theta(\boldsymbol{Z}^i)\right) \leq \sqrt{2 - 2\left(1 - \frac{2k}{n+k}\right)^{\frac{k}{2}}} \tag{28}$$

Now, we analyze the behavior of this bound. We rewrite the upper bound as follows:

$$\sqrt{2} \cdot \sqrt{1 - \left(1 - \frac{2k}{n+k}\right)^{\frac{k}{2}}}$$

We first define $p := \frac{k}{n}$. Then we have:

$$1 - \frac{2k}{n+k} = \frac{n-k}{n+k} = \frac{1-p}{1+p}$$

By Taylor series, we have the following series expansion:

$$\ln\left(\frac{1-p}{1+p}\right) = \ln(1-p) - \ln(1+p) = -p - \frac{p^2}{2} - \frac{p^3}{3} + \cdots - \left(p - \frac{p^2}{2} + \frac{p^3}{3} + \cdots\right)$$

$$= -2p - \frac{2}{3}p^3 + \cdots \tag{29}$$

Then we have the following behavior:

$$1 - \frac{2k}{n+k} = \left(\frac{1-p}{1+p}\right)^{\frac{k}{2}} = \exp\left(\frac{k}{2}\ln\left(\frac{1-p}{1+p}\right)\right)$$

$$= \exp\left(\frac{k}{2}\left(-2p - \frac{2}{3}p^3 + \cdots\right)\right)$$

$$\sim \exp\left(-kp - \frac{k}{3}kp^3\right) \tag{30}$$

$$\sim \exp\left(-\frac{k^2}{n}\right)$$

Thus, we have:

$$D_{\mathrm{TV}}\left(\boldsymbol{S}_\theta(\boldsymbol{Z}) \parallel \boldsymbol{S}_\theta(\boldsymbol{Z}^i)\right) \leq \sqrt{2 - 2\left(1 - \frac{2k}{n+k}\right)^{\frac{k}{2}}} \sim \mathcal{O}\left(\sqrt{1 - \exp\left(-\frac{k^2}{n}\right)}\right) \tag{31}$$

This behavior indicates that the bounds gets weaker exponentially with larger $k$.

**Remark 2** (Measuring $k$). *The parameter $k$ in the bound can be measured empirically for each video calibration dataset. To measure $k$, for each of the three test datasets in Table 3, we measure the average changes of labels of subjects/activities after swapping the $i$-th data point with the last one. This value is usually low (in most cases $\leq 2$).*

## D. Training and Testing Datasets Details

We use the standard 3D human shape-pose datasets: 3DPW (Von Marcard et al., 2018), Human3.6M (Ionescu et al., 2013), MPII-3DHP (Mehta et al., 2017), Penn Action (Zhang et al., 2013), PoseTrack (Andriluka et al., 2018), and InstaVariety (Kanazawa et al., 2019) where the preprocessed data is provided by (Shen et al., 2023), (Choi et al., 2021), and (Kocabas et al., 2020), and evaluated on 3DPW, Human3.6M, MPII-3DHP. Note that our training dataset is about 2.5% smaller than previous works because we hold out a small portion ($\sim$ 1500 datapoints) for calibration.

# E. Details of Choice of Conformity Score Function

Note that all three mentioned conformity score functions were trained end-to-end with the training time ensemble augmentation setting since learning the score function after the human reconstructor is trained does not improve the performance. For the score function augmented with inefficiency loss, we are essentially controlling the size of the conformal prediction set during training (Stutz et al., 2021). For the classifier-style conformity score function, the training objective is to classify if the mean 3D keypoints L2 loss is within 40mm from the groundtruth using BCE loss and we use the logit as the conformity score. It is worth noting that all three variants result in better performance, demonstrating the importance of training time ensemble augmentation. Note that Ineff. needs more proposals during training than others and converges more slowly.

# F. Test-Time Multi-Hypothesis Aggregation

In the above experiments, we compare single-hypothesis outputs across methods for fairness, as many baselines are single-output by design. MC Dropout is optional and provides two key advantages: (1) it naturally yields multiple hypotheses, and (2) using conformity scores, we can aggregate them via a weighted average that suppresses low-quality predictions based on the DUF score value.

To clarify its benefit, we provide a comparison below showing how multi-hypothesis (H=20, cutting off samples below calibrated threshold) aggregation improves performance over the single-sample case.

Table 4: Improvement on 3DPW

|  | MPJPE | MPVPE | Accel | PA-MPJPE |
|---|---|---|---|---|
| Improvement | -0.9 | -1.2 | -1.1 | -1.1 |

Table 5: Improvement on MPI-INF-3DHP

|  | MPJPE | Accel | PA-MPJPE |
|---|---|---|---|
| Improvement | -2.3 | -0.1 | -1.6 |

Table 6: Improvement on Humans.6M

|  | MPJPE | Accel |
|---|---|---|
| Improvement | -2.2 | -0.2 |

As we can see, with multi-hypothesis aggregation, the results get further improved. The GPU usage is under 12 GB.

# G. Performance under Occlusion

Regarding occlusion/truncation tests, we have completed larger-scale quantitative experiments. We rerun Table 1 experiments (using the same trained model), but with all input image sequences' bottom 25% truncated.

Table 7: Performance comparison with and without truncation

| Method | 3DPW | | | | MPI-INF-3DHP | | | Human3.6M | |
|---|---|---|---|---|---|---|---|---|---|
|  | PA-MPJPE | MPJPE | MPVPE | Accel | PA-MPJPE | MPJPE | Accel | PA-MPJPE | MPJPE |
| W/o Truncation | 48.7 | 76.2 | 91.7 | 6.9 | 61.3 | 92.8 | 7.2 | 44.0 | 63.8 |
| W/ Truncation | 50.7 | 79.7 | 94.8 | 7.0 | 62.1 | 93.2 | 7.8 | 46.1 | 64.9 |

Despite being 25% truncated, the performance loss is not much, and in many cases, still better than the untruncated baselines in Table 1, indicating CUPS's robustness to occlusion.

## H. Limitations

While CUPS performs well on various benchmarks, we acknowledge that it does have some limitations. First, many samples need to be proposed during training to improve the learned nonconformity score, which consumes a lot more GPU memory (30% more going from 10 proposals to 20) and slows down the training process. Second, the method does not utilize human joint-level information, which could potentially improve the performance.

