# OpenReview forum: "CUPS: Improving Human Pose-Shape Estimators with Conformalized Deep Uncertainty"
_ICML.cc/2025/Conference — ICML 2025 poster_

### Official Review · Reviewer_wgxo · 2025-03-13

**Overall Recommendation:** 3

**Summary:**

The paper introduces CUPS, a video-based HMR approach with uncertainty quantification. Specifically, the method uses GLoT as a base model to extract global and local features from videos. An adversarial loss is defined on the output meshes during training, similar to VIBE. The discriminator output (from a sigmoid layer) estimates the uncertainty of the predictions (whether they belong to the real dataset or not). Next, the paper applies the statistical tools developed by Barber et al. to calibrate the uncertainties. The paper presents two theoretical bounds and demonstrates state-of-the-art performance on standard benchmarks.

**Claims And Evidence:**

The claims are generally supported, including achieving state-of-the-art and improved generalizability. The proofs also follow prior references.

**Essential References Not Discussed:**

All references in the manuscript are adequate. However, I am interested in the authors' opinions on other uncertainty prediction approaches, like RLE [1] or similar works, and how they could be adapted into their framework.

[1] Li, Jiefeng, et al. "Human pose regression with residual log-likelihood estimation." Proceedings of the IEEE/CVF international conference on computer vision. 2021.

**Experimental Designs Or Analyses:**

All experiments were reviewed, and they are sound.

**Methods And Evaluation Criteria:**

The paper uses a recent paper (GLoT) as a base model, which is appropriate. Additionally, it tests the model on popular benchmarks like 3DPW and Human3.6m, according to prior research. The evaluation metrics also follow the standards in this field.

**Other Comments Or Suggestions:**

I suggest including the values of hyperparameters, conformal dataset size, and architectural details in the manuscript or supplementary materials. Please see my other comments for more suggestions/questions.

**Other Strengths And Weaknesses:**

### Strengths
1. The paper provides a creative combination of existing ideas, resulting in improvements.
2. The paper is well-written.

### Weaknesses
1. The paper does not fully explore the limitations of calibration methods, which often involve memory and computation costs. More info on this (more than supplementary) would be appreciated.
2. Some parts are unnecessarily explained, especially where it is not the paper's main contribution. Given the recency of the tools, it could be acceptable.
3. No comparisons are provided with other uncertainty modeling papers.
4. The results of the method are not very surprising, as adding any uncertainty-aware modules may result in better performance.
5. The contributions are somewhat limited, especially given similar works like [1]. The paper's main focus, uncertainty modeling, and calibration is not experimented upon, and there are minimal comparisons with other approaches.

References:
[1] Zhang, Harry, and Luca Carlone. "CHAMP: Conformalized 3D Human Multi-Hypothesis Pose Estimators." arXiv preprint arXiv:2407.06141 (2024).

**Questions For Authors:**

1. Could you include the hyperparameter's chosen values in the manuscript or supplementary materials?
2. How impactful is the choice of feature distance function? How important is the choice of the temperature? And did you consider using rotational distance instead of L2 distance of rotations?
3. Why is the beta defined with different values across frames? Specifically, in the problem formulation in line 160 (left), you define a separate beta for each frame. If it represents the shape parameters, should it not be the same for all frames?
4. I understand if it is not possible to incorporate other uncertainty estimation approaches and provide their results, but could you elaborate and discuss how they could be implemented into your pipeline? and if it would be an appropriate choice?

**Relation To Broader Scientific Literature:**

Uncertainty quantification and conformal prediction are valuable tools for machine learning tasks, especially for HMR, where a large portion of the SMPL parameter space does not represent a plausible pose. As a result, obtaining the uncertainty and calibrating it can significantly improve the estimation performance. Such approaches are often found in medical papers, where uncertainty is paramount. However, more recent papers have adapted uncertainty estimation into HMR, showing significant performance gains. Therefore, given the context of recent research, this paper is timely and addresses an interesting topic.

**Theoretical Claims:**

All proofs are according to prior research from Barber et al. 2023.

---

> ### Author Rebuttal · Authors · 2025-03-31
>
> We sincerely thank Reviewer wgxo for their thoughtful feedback and insightful questions. We especially appreciate their recognition of our **creativity** and **theoretical soundness**. Below, we address their comments in detail.
>
> > On calibration’s computational cost:
>
> Calibration in CUPS is **lightweight**. We hold out a small subset (<1000 samples) from the training data as a calibration set. In practice, we only use 500 calibration samples (i.e., about 8 forward-pass batches), and once the conformity threshold is computed, it remains fixed. No additional inference-time cost is incurred.
>
> > On hyperparameters and distance function:
>
> Thank you for pointing this out. We **will include a full list of hyperparameters** in the appendix. The missing one now is the **temperature**, set to 20. While both temperature and the ablated parameter $\lambda$ affect score loss strength, $\lambda$ has a greater impact.
>
> Regarding the distance function, since we operate in high-dimensional feature space, rotation-based metrics are inapplicable. An alternative we explored was using Cosine similarity, which also works well.
>
> > On alternative uncertainty methods and adapting other backbones:
>
> We appreciate the reference to RLE [1]. RLE formulates pose estimation as distribution matching, embedding uncertainty into the architecture via flow matching. In contrast, CUPS uses MC Dropout to emulate a probabilistic output space but remains modular—*any probabilistic/generative backbone like RLE could be used within the Conformal Prediction (CP) framework*.
>
> CP’s model-agnostic nature allows CUPS to **adapt to new backbones with minimal changes while maintaining theoretical coverage** guarantees. For example, across all metrics, CUPS outperforms Dwivedi et al. (2024), which uses learned occlusion confidences for pose uncertainty measurement but lacks shape uncertainty modeling and theoretical coverage guarantees. Unlike methods that inject robustness via heavy architectural modifications and constraints, CUPS provides **unified, reliable uncertainty quantification for both pose and shape** with minimal overhead from conformal prediction.
>
> Thus, when it comes to selecting other models for our pipeline, we would *still use conformal prediction* but maybe incorporate probabilistic (generative) models, which give multi-hypothesis outputs efficiently.
>
> > On comparison with CHAMP:
>
> Thank you for highlighting this. We cite CHAMP in Section 2, as it inspired CUPS. However, CUPS addresses key limitations noted in CHAMP’s paper:
>
> - Pose-only: CHAMP focuses solely on pose estimation, whereas CUPS extends conformal prediction to pose-shape models, enabling richer and more expressive representations of human motion.
>
> - Theoretical rigor: CHAMP’s application of CP to **non-exchangeable** datasets like human motion videos leads to mostly empirical coverage guarantees since the CP assumptions are violated. CUPS builds on recent advances in CP **beyond exchangeability**, providing a **rigorous theoretical framework** that accounts for the structure and characteristics of video-based datasets.
>
> > On shape parameter consistency (Line 160):
>
> We apologize for any confusion we might have caused. The reviewer is right in pointing out that the shape parameters should remain consistent across frames, but for the sake of **mathematical formulation**, especially with the use of a sequential transformer model, the pose and shape parameters are outputted as a **sequence**. While we do not enforce each frame’s shape to be exactly the same, empirical outputs show consistent shapes across frames, and it might be interesting to **explore losses that regularize shape consistency**.
>
> > Misc.
>
> We thank the reviewer for pointing out the redundancy in some introduction of the methods. We will make our writing more concise.
>
> The main uncertainty-based baselines we compare with are Dwivedi et al. (2024) and 3DMB (Biggs et al., 2020), which we outperformed by a noticeable margin. Other uncertainty-based baselines, which we are exploring right now, include swapping out GLoT with different probabilistic HMR models.
>
> Adding uncertainty itself **does not necessarily improve the results**. Our main contribution is a framework that quantifies the uncertainty with theoretical rigor while backpropagating the uncertainty into the pose learning process with delicate designs that improve the final outputs. Most UQ methods such as CP do not "close the loop"; they assume a well-trained model, and **very few prior works have explored incorporating uncertainty into the learning process.** As Reviewer aBRZ said, CUPS "offers novel synergies to advance **safety critical** vision systems."

---

> > ### Comment · Reviewer_wgxo · 2025-04-02
> >
> > I thank the authors for their detailed rebuttal and for addressing my questions, particularly regarding the RLE and its incorporation into the CP framework. These clarifications have resolved my primary concerns. I have no further questions at this time. I am revising my recommendation after considering the authors' responses and the other reviews. I recommend Minor Acceptance, conditional upon incorporating these clarifications into the manuscript or supplementary materials. This work demonstrates sufficient novelty for video HMR within the context of this conference and offers a valuable foundation for future research in this area.

---

> > > ### Author Response · Authors · 2025-04-02
> > >
> > > We sincerely appreciate Reviewer wgxo for raising the score and for acknowledging that CUPS **offers a valuable foundation for future research**. We will make sure to incorporate the comments and clarifications into the final version.

---

### Official Review · Reviewer_aBRZ · 2025-03-14

**Overall Recommendation:** 3

**Summary:**

This paper introduces CUPS, a method that integrates conformal prediction with deep uncertainty learning for 3D human pose-shape estimation from monocular videos. The key innovation lies in training an end-to-end deep uncertainty function alongside the reconstruction model, which serves as a conformity score for constructing prediction sets with statistical guarantees. By addressing non-exchangeability in video data through weighted conformal calibration, CUPS achieves state-of-the-art performance on benchmarks, while providing theoretical bounds on coverage gaps. The method is validated through extensive experiments, ablation studies, and in-the-wild tests.

## update after rebuttal

As the author addresses most of my and others' concerns, esp. for the last clarification on the Multi-Hypo, alignment and additional quantitative truncation test, I decided to keep my recommendation of Weak Accept. It would be interesting to also test the method on a more powerful backbone.

**Claims And Evidence:**

The claims are well-supported:
- Good performance is evidenced by quantitative comparisons. The effectiveness of uncertainty-aware training is demonstrated via ablation studies (e.g., training-time ensemble augmentation)
- Some mathematical proofs are provided to support the correctness of the theorems. Theoretical coverage guarantees are derived for non-exchangeable data, with empirical validation. Minor improvements could include clarifying the practical implications of the coverage bounds.

**Essential References Not Discussed:**

To my knowledge, the paper cites relevant essential works. A few literature could also be discussed, including probabilistic multi-hypo methods (MHEntropy, ICCV'23) and uncertainty HPE (though 2D, PlausibleUncertainties DER, ICCV'23).

**Ethical Review Concerns:**

NA.

**Experimental Designs Or Analyses:**

Experiments are comprehensive, covering some datasets and baselines and ablation studies.

May I ask:
- **(Comparisons with multi-hypo methods)** The authors seem not to clarify why they discuss but do not compare with multi-hypo aggregation methods?
- **(Figs. 1 & 6 multi hypos)** seem not so meaningful as like hands are not aligned with the clear image? I would expect to see they span along the depth from the side view.
- **(GAN DUF)** Introducing adversarial loss is indeed meaningful, but could the instability of adversarial training potentially led to difficulties in model convergence? Additionally, to what extent does adversarial loss impact the results? I am not talking about $\lambda$ in Fig. 5 but GAN training setting and hyperparams. Corresponding ablation experiments can be conducted to explore these questions.
- **(CP under occlusion)** Could analysis be done 3DPW-occlusion (like PARE), 3DPW-truncation (like NIKI)? What will the result change to?

**Methods And Evaluation Criteria:**

- The methodology is sound: the global-local transformer architecture aligns with recent advances, and the integration of adversarial training for uncertainty scoring seems interesting.
- Evaluation metrics (MPJPE, PA-MPJPE, Accel) and datasets are standard for 3D pose estimation. The inclusion of in-the-wild tests strengthens practical relevance.

May I ask:
- **(training & inference)** Is it true that augmenting multiple training samples $H$ during the training but only doing **SINGLE** inference during test on standard benchmarks (e.g. Tab. 1)? MCDropout multi-hypo generation and set selection according to the DUF is an optional function? Or inference also samples several and aggregates them with the DUF.
- **(Degenerated discriminator)** GAN studies find the goal of discriminator is to help the learning of the generator and it will eventually degrade. However, it seems discriminator in the paper works well. Could the author elaborate and provide some insights?

**Other Comments Or Suggestions:**

Please see other sections.

**Other Strengths And Weaknesses:**

**[Strengths]**
- Reasonable integration of conformal uncertainty learning and multi-hypo HMR.
- Superior empirical results across datasets.
- also provides some practical contributions (e.g., MC dropout seems to well suit BERT-like masking for DUCS construction).

**[Weaknesses]**
- Limited discussion on **computational overhead** as multi-hypo DUCS prediction.
- **(Corner hard cases)** The model's ability to quantify uncertainty under extreme scenarios like severe occlusions or rapid motion is crucial. While tested on in-the-wild videos, the paper does not explicitly evaluate performance under these situations.

**Questions For Authors:**

Please see the above.

**Relation To Broader Scientific Literature:**

This work bridges three key research threads: conformal prediction, 3D human pose-shape estimation, and deep uncertainty quantification, offering novel synergies to advance safety critical vision systems.

**Theoretical Claims:**

To my perspective, the proofs for Theorems 1–3 are logically structured. The adaptation of Barber et al.’s framework to handle non-exchangeable data is appropriate.

- **($\beta$ distribution)** Theorem 3 assumes that the conformity scores follow a Beta distribution. How to validate this assumption in practice?

---

> ### Author Rebuttal · Authors · 2025-03-31
>
> We sincerely thank Reviewer aBRZ for their thoughtful feedback and insightful questions. We especially appreciate their recognition of the **strength of our experiments** and **practical and theoretical contributions,** offering **novel synergies to advance safety critical vision systems**. Below, we address their comments in detail.
>
> > On training augmentation and multi-hypothesis aggregation:
>
> The reviewer is correct: we compare single-hypothesis outputs across methods for **fairness**, as many baselines are single-output by design. MC Dropout is optional and provides two key advantages: (1) it naturally yields multiple hypotheses, and (2) using conformity scores, we can aggregate them via a weighted average that suppresses low-quality predictions: $ \bar{x} = \sum_i w_i \cdot x_i, \text{ where } w_i \sim \text{conformity score}(x_i)$
>
> To clarify its benefit, we provide a comparison below showing how multi-hypothesis (H=20, cutting off samples below calibrated threshold) aggregation improves performance over the single-sample case.
>
> |  | **Improvement on 3DPW** |        |        |        | **Improvement on MPI-INF-3DHP** |        |        |      **Improvement on Human3.6M** |        |        |
> |-----------|--------------------------|--------|--------|--------|-----------------------------|--------|------------|----------------------------|--------|--------|
> |           | PA-MPJPE               | MPJPE | MPVPE | Accel | PA-MPJPE                | MPJPE| Accel | PA-MPJPE    | MPJPE | Accel |
> |           | -0.9  | -1.2 | -1.1 | -0.2 | -1.1   | -2.3 | -0.1 |-1.6 |  -2.2  | -0.2 |
>
> As we can see, with multi-hypothesis aggregation, the results get further improved. The GPU usage is under 12 GB.
>
> > On the score function and hyperparameters:
>
> While the score function resembles a discriminator, it is fundamentally different from a GAN setup. Its role is not to classify real/fake samples but to **rank predictions** based on conformity. We use a weaker adversarial loss, and backpropagate the score loss only once every 100 SMPL updates to ensure training stability—more frequent updates (e.g., every 10–50 steps) lead to instability, as we will discuss in the final version.
>
> MC Dropout also induces **distributional oscillations** during training, which can prevent full convergence of the score function. This is acceptable: conformal prediction requires only that the conformity score be consistent across calibration samples, not fully converged.
>
> > On the choice of the beta distribution:
>
> We chose the beta distribution based on empirical evidence:
>
> - Calibration scores fitted well via MLE. Please checkout this [new link](https://sites.google.com/view/cups-occlusion-supp/home) for **plots of fitted vs. theoretical Beta alignment**.
> - The Kolmogorov–Smirnov test failed to reject the beta hypothesis.
>
> This choice allows for analytical tractability in deriving theoretical bounds while remaining grounded in the observed data distribution.
>
> > On hard OOD examples and occlusions:
>
> Some in-the-wild videos we show are fast-paced and more challenging than the training data. While a few examples in the paper may not illustrate this well, we encourage the reviewer to visit [our website](https://sites.google.com/view/champpp) for full videos. These show **more diversity in foot/hand motion and depth variation**. We also add **two new visualizations** highlighting CUPS’s robustness under heavy occlusions in a separate [anonymous website](https://sites.google.com/view/cups-occlusion-supp/home).
>
> CUPS does occasionally inherit failure cases from its GLoT backbone, but it requires only minimal architectural changes. Importantly, since CP is **model-agnostic**, stronger backbones can be **easily swapped** in for future improvements while preserving theoretical guarantees.
>
> > On missing related work:
>
> We thank the reviewer for flagging the two missing works:
>
> MHEntropy introduces entropy-based methods for hand pose-shape recovery, especially effective under occlusion. As discussed above, while CUPS does not explicitly model occlusions (we assume the underlying backbone is somewhat robust), we could further improve CUPS by incentivizing diverse outputs during training via entropy regularization. This could potentially yield more robust and diverse outputs.
>
> Plausible Uncertainties focuses on 2D pose regression. While useful, our choice of Conformal Prediction offers broader applicability, minimal architectural modification, and theoretical guarantees, making it better suited for general 3D human pose-shape tasks.
>
> > On set prediction’s computation cost:
>
> Multi-hypo prediction in CUPS is **lightweight**. We are able to achieve, on average, 20ms/segment on video data on a single V100 GPU. This is because MC Dropout itself is not memory-expensive. Other approaches, such as diffusion-based probabilistic backbones, might consume more memory, but the bottleneck does not come from CP or calibration.

---

> > ### Comment · Reviewer_aBRZ · 2025-04-04
> >
> > I sincerely thank the author for the careful feedback. There are still some concerns I want to clarify:
> > - Multi-hypo methods do not mean the output is all multiple. They also report the result of a single aggregated prediction like D3DP. Yet, most of them are skeleton-based and have different settings.
> > - For multi hypos, I still find the quality is unsatisfactory as there are many cases where the hypotheses are not aligned with the image and show unwanted diversity.
> > - For occlusion/truncation sensitivity test, it would be encouraged to include quantitative results instead of just qualitative ones.
> >
> > I would likely keep my scores and see other review discussions. Thanks.

---

> > > ### Author Response · Authors · 2025-04-04
> > >
> > > We sincerely appreciate Reviewer aBRZ for their feedback and for acknowledging the effort of our reply.
> > >
> > > - Regarding multi-hypo fairness, we apologize for any confusion we might have caused. We do indeed aggregate the multiple hypotheses, and the performance gain above is achieved by this aggregation. We did not put this in the original manuscript because it is unclear if this aggregation is fair to single-hypothesis methods. However, we will add it to the final version.
> > >
> > > - Regarding some unsatisfactory results, we acknowledge that some results are off due to the backbone we are using. For many test cases from OOD in-the-wild videos, using just the backbone model, GLoT, the alignment was worse off -- CUPS actually **improved the alignment** here with the E2E conformity score function. While CUPS's score function improves the backbone performance **by a noticeable margin**, the improvement will not be unbounded. However, CUPS has two key properties that alleviate this issue:
> > >   - CUPS is **modular**, which gives us a chance to swap out the backbone for better ones in order to achieve better alignment results. This involves minimal framework change, and we are working on incorporating a diffusion-based model as CUPS's backbone.
> > >   - CUPS's DUCS ranks and filters out "bad" hypotheses. One of our main contributions is the ability to **score the outputs with mathematical guarantees**. While some outputs are off (i.e., unwanted diversity), they will be downweighted during the aggregation step, potentially **reducing the impact of bad outputs.**
> > >
> > > - Regarding occlusion/truncation tests, we have completed larger-scale quantitative experiments. We rerun Table 1 experiments (using the same trained model), but with all input image sequences' bottom 25% truncated.
> > >
> > > | Method                | 3DPW |          |          |          | MPI-INF-3DHP |          |          | Human3.6M |          |
> > > |-----------------------|----------------------------------|----------|----------|----------|----------------------------------|----------|----------|----------------------------------|----------|
> > > |                       | PA-MPJPE | MPJPE | MPVPE  | Accel | PA-MPJPE | MPJPE | Accel | PA-MPJPE | MPJPE | Accel |
> > > | **W/o Truncation**              | 48.7   | 76.2 | 91.7 | 6.9      | 61.3   | 92.8| 7.2  | 44.0   | 63.8 | 3.5 |
> > > | **W/ Truncation**              | 50.7  | 79.7 | 94.8 | 7.0      | 62.1   | 93.2 | 7.8  | 46.1   | 64.9 | 3.7 |
> > >
> > > Despite being 25% truncated, the performance loss is not much, and in many cases, still better than the untruncated baselines in Table 1, indicating CUPS's **robustness** to occlusion. We will try more datasets that Reviewer aBRZ mentioned, such as 3DPW-Occlude and 3DPW-Truncate, and incorporate the results in the final version.
> > >
> > > We hope our answers have further clarified your concerns and questions. Please let us know if you have further comments. Thanks.

---

### Official Review · Reviewer_Xj5H · 2025-03-14

**Overall Recommendation:** 3

**Summary:**

This paper presents CUPS, an approach to infer 3D human shapes and poses from videos. At the core is a deep uncertainty function that is trained with 3D pose estimation, and it computes a conformity score to optimize the pose prediction in inference. Experimental results on different datasets and metrics demonstrate that the proposed method outperforms existing baseline methods on human pose estimation tasks.

**Claims And Evidence:**

This paper conducts experiments on different datasets including 3DPW, MPI-INF-3DHP, and Human3.6M, and the experimental results illustrate the effectiveness of the the approach.

**Essential References Not Discussed:**

The references are good.

**Experimental Designs Or Analyses:**

The experimental analysis is sound.

This paper conducts experiments on different datasets including in-the-wild videos, and also provides experimental analysis (e.g., Empirical Coverage) to verify the proposed paper.

**Methods And Evaluation Criteria:**

The method was evaluated on different metrics including PA-MPJPE, MPJPE, and MPVPE. The results show that CUPS outperforms existing methods.

**Other Comments Or Suggestions:**

Cite and discuss CHAMP: Conformalized 3D Human Multi-Hypothesis Pose Estimators.

**Other Strengths And Weaknesses:**

Strength:
This paper is well-written and easy to follow.
The paper solves an interesting problem of 3D human pose and shape prediction from videos.


Weakness:
Some components in the paper are not thoroughly verified.
For example, how do the global and local transformer improve the results?
How to define the global and local features, and how are they decoupled?

**Questions For Authors:**

How does the deep uncertainty function decouple pose and shape for motion correction? For example, we can adjust the pose or shape to make the prediction aligned with the ground truth in training.

What’s the advantage of the deep uncertainty function over the pose discriminator used in VIBE?

**Relation To Broader Scientific Literature:**

This paper discussed the relations to existing methods in Related Work.

**Theoretical Claims:**

I have checked the Definition and Theorem 1-12, which are technically solid.

---

> ### Author Rebuttal · Authors · 2025-03-31
>
> We sincerely thank Reviewer Xj5H for their thoughtful feedback and insightful questions. We especially appreciate their recognition of the **strength of our experiments**, the **theoretical contributions** of our work, and the **effectiveness** of the approach. Below, we address each of the reviewer’s comments in detail.
>
> > On comparison with CHAMP: Conformalized 3D Human Multi-Hypothesis Pose Estimators
>
> Thank you for highlighting this connection. We are indeed aware of CHAMP and have cited it in Section 2. CHAMP served as a key inspiration for our work, as it is among the few papers that explore conformal prediction for human pose estimation. However, CUPS addresses two major limitations noted in CHAMP:
>
> - Pose only: CHAMP focuses solely on pose estimation, whereas CUPS extends conformal prediction to pose-shape models, enabling richer and more expressive representations of human motion.
>
> - Theoretical soundness: CHAMP’s application of CP to **non-exchangeable** datasets like human motion videos leads to mostly empirical coverage guarantees since the **CP assumptions are violated**. On the other hand, CUPS builds on recent advances in **CP beyond exchangeability**, providing *a rigorous theoretical framework* that accounts for the structure and characteristics of video-based datasets.
>
> > On the global and local transformer components:
>
> We apologize for any confusion. As cited in the paper, CUPS builds upon the GLoT backbone (Shen et al., 2023), which introduced the global and local transformer modules. While these are not novel contributions of CUPS (and this should be correctly represented in the paper), we summarize their roles here for clarity:
>
> - The global transformer captures *long-range temporal dependencies* to ensure consistency in human motion across frames.
>
> - The local transformer focuses on *fine-grained temporal dynamics*, refining predictions by modeling short-term variations around mid-frames.
>
> The combination of these modules produces a decoupled global-local representation. For implementation details, please refer to GLoT’s [official codebase link](https://github.com/sxl142/GLoT/blob/main/lib/models/GLoT.py#L53).
>
> > On the deep uncertainty function (DUF) for pose and shape:
>
> CUPS applies conformal prediction to the output SMPL parameters, thereby **quantifying uncertainty over both pose and shape**. The learned conformity score function *implicitly* decouples the two: when either pose or shape is off, the conformity score may be low; when only one is off, the score may still be high depending on the calibration distribution.
>
> While it is possible to explicitly decouple pose and shape by using two separate conformity scores (and thus two CP procedures), doing so would complicate the theoretical analysis. Modeling the *interdependencies* between pose and shape would be necessary to maintain valid performance bounds.
>
> > On comparison with VIBE’s discriminator:
>
> Thank you for raising this point. While both VIBE’s motion discriminator and CUPS’s DUF are trained adversarially, they serve distinct roles:
>
> - VIBE’s discriminator relies on **motion priors** from AMASS, introducing supervision from external datasets.
>
> - CUPS’s DUF, in contrast, is **self-supervised**. It is trained using an ensemble of predictions generated via Monte Carlo dropout, requiring no additional data.
>
> This self-supervised setup makes CUPS **more modular and efficient**, introducing minimal changes to the backbone architecture. Moreover, because CP is model-agnostic, CUPS can be easily adapted to other backbones beyond GLoT while retaining its theoretical guarantees.

---

### Official Review · Reviewer_Xcn3 · 2025-03-15

**Overall Recommendation:** 3

**Summary:**

This paper introduces a novel method for human pose and shape estimation, utilizing the SMPL representation, from video sequences. The proposed approach incorporates conformalized deep uncertainty modeling, which allows for the generation of multiple samples, in contrast to the single-output methods commonly found in the literature. The uncertainty is theoretically calibrated, providing a safety guarantee for robotics and other downstream applications. The experiments are comprehensive and demonstrate state-of-the-art performance compared to existing baselines.

**Claims And Evidence:**

The authors claim to propose a new method that provides both human pose and shape, along with the conformalized uncertainty scores. These claims are supported by detailed methods and extensive experiments demonstrating the performance.

**Essential References Not Discussed:**

N/A

**Experimental Designs Or Analyses:**

The experiments are extensive, and the results are impressive. Under uncertainty modeling, the method achieves state-of-the-art performance across nearly all metrics when compared to the baselines.

**Methods And Evaluation Criteria:**

The high-level approach involves a transformer-based architecture that predicts the parameters of the SMPL representation, utilizing a global regressor and a local corrector that focuses on different parts of the video sequences.

The uncertainty is predicted by training a neural network that takes both the input X and the output Y, providing a probability score between 0 and 1. This design is similar to a discriminator, with the authors employing a similar loss function to supervise the network.

The authors also address the important issue of "calibration" to ensure that the output uncertainty score is well-calibrated within a probabilistic framework. Detailed and reasonable proofs are provided both in the main text and the supplementary material.

**Other Comments Or Suggestions:**

N/A

**Other Strengths And Weaknesses:**

Uncertainty modeling is a highly valuable yet underexplored area in the literature. Typically, we focus on achieving high scores while overlooking the inherent ambiguity within neural networks. This method offers an additional measure to obtain robust outputs and certification with uncertainty, which is an important contribution to the community.

**Questions For Authors:**

1. The multiple samples seem to lack diversity. For instance, in Figure 6, there is no sample covering the foot. What is the main reason for this, and how can this issue be addressed to improve the method further?

2. The motivation for using conformal prediction to model uncertainty is not entirely clear. This method requires exchangeable input data. While the related work discusses alternative methods that use, for example, explicit confidence values, the authors do not provide a summary of their limitations or how this paper's method stands out. What led the authors to choose this particular uncertainty modeling approach over others?

**Relation To Broader Scientific Literature:**

A human pose and shape estimator with uncertainty scores can be applied in safety-critical areas, such as robotics, enabling a wide range of applications.

**Theoretical Claims:**

The architecture of the network follows standard deep learning practices, such as Transformers and MLPs. The main theoretical contribution lies in the calibration of uncertainty, which is sound and well-justified.

---

> ### Author Rebuttal · Authors · 2025-03-31
>
> We sincerely thank Reviewer Xcn3 for their thoughtful feedback and insightful questions. We particularly appreciate their recognition of the **theoretical soundness** of our work, **the need for uncertainty prediction in human pose estimation in the community**, and the applicability to **safety-critical areas**. Below, we respond to their comments in detail.
>
> > On the lack of diversity in Figure 6:
>
> We appreciate the reviewer’s observation. CUPS was trained on multiple human pose-shape datasets, many of which, such as Human3.6M, were collected in controlled indoor environments. This may limit the diversity of foot and lower-body movements represented during training. The bottom example in Figure 6 depicts an outlier instance with highly dynamic motion that is not well-represented in the training set. Despite this, CUPS performs reasonably well, **tracking both pose and shape in such out-of-distribution (OOD) scenarios**.
>
> We note that Figure 6 presents only static snapshots. We encourage the reviewer to explore [our website](https://sites.google.com/view/champpp), where full prediction videos are available. These showcase significantly **more diversity in predicted meshes**—including many cases where foot motion is well captured.
>
> To improve diversity further, two directions are promising: (1) Incorporating more in-the-wild video data during training to reduce reliance on constrained datasets like Human3.6M, and (2) Introducing entropy regularization into the output ensemble [1], encouraging greater variability in the predictions.
>
> > On our choice of the conformal prediction (CP) framework:
>
> This is an excellent question. We selected CP due to its **flexibility and generalizability**. CP is a distribution-free framework for uncertainty quantification that can be applied to any machine learning model (Angelopoulos & Bates, 2021; Shafer & Vovk, 2008). Crucially, even without strict exchangeability (*as is the case with human video data*), CP enables us to estimate a theoretical lower bound on performance by leveraging the properties of the calibration dataset.
>
> This has two key advantages:
>
> - Theoretical guarantees: CP allows us to offer probabilistic performance guarantees via conformity scores—effectively certifying the reliability of predictions.
>
> - Modularity: CUPS requires only minimal changes to the backbone architecture. Because CP is model-agnostic, one could replace the GLoT-based backbone with other architectures in the future to improve accuracy, while still benefiting from CUPS’s theoretical guarantees.
>
> Alternative approaches, such as Dwivedi et al. (2024), introduce learned occlusion confidences but lack the theoretical coverage guarantees that CP offers. Moreover, Dwivedi et al. (2024)’s method is only applied to pose, while uncertainty quantification for shape estimation remains unexplored. CUPS provides a unified method for uncertainty quantification for both pose and shape since CP is used in SMPL space. Other prior work relies on injecting robustness via additional constraints, which often demands *substantial modifications* to the prediction pipeline. In contrast, CUPS offers **robust and theoretically grounded predictions with minimal architectural overhead**.
>
>
>
>
>
> [1] Chen, Rongyu, Linlin Yang, and Angela Yao. "Mhentropy: Entropy meets multiple hypotheses for pose and shape recovery." Proceedings of the IEEE/CVF International Conference on Computer Vision. 2023.

---

> > ### Comment · Reviewer_Xcn3 · 2025-04-06
> >
> > I would like to thank the authors for providing a detailed explanation to address the questions. I have no further questions and would like to keep my scores leaning toward acceptance.

---

> > > ### Author Response · Authors · 2025-04-06
> > >
> > > We sincerely appreciate Reviewer Xcn3 for their acknowledgement of our contributions and recommendation of acceptance. We will incorporate the discussed points in our final version.

---

### Decision · Program_Chairs · 2025-05-01

**Decision:**

Accept (poster)

**Comment:**

This paper introduces a novel method for learning 3D human shapes and poses from RGB videos with built-in uncertainty quantification using conformal prediction. By generating and scoring multiple hypotheses during training, the method integrates a deep uncertainty function trained jointly with the pose estimator. This enables conformal prediction with theoretical coverage bounds, leading to strong performance with probabilistic guarantees.

The paper was reviewed by four experts, and all reviewers unanimously support acceptance of the paper. While there were some concerns initially around the limitations of calibration methods, need for comparisons with other uncertainty modeling work, as well as some questions on the stability of the proposed method, the rebuttal helped answer these questions, and reviewers (Rev#wgxo) appreciated the thoughtful responses and the value of this method for video HMR.

Given the consensus, this paper is recommended for acceptance. As noted by reviewers, the authors are recommended to incorporate the rebuttal responses in the final version.